# LICO: Explainable Models with Language-Image COnsistency

**Yiming Lei**[1], **Zilong Li**[1], **Yangyang Li**[2], **Junping Zhang**[1], **Hongming Shan**[3,4*]

[1] Shanghai Key Laboratory of Intelligent Information Processing,
School of Computer Science, Fudan University
[2] Academy of Mathematics and Systems Science, Chinese Academy of Sciences
[3] Institute of Science and Technology for Brain-Inspired Intelligence and
MOE Frontiers Center for Brain Science, Fudan University
[4] Shanghai Center for Brain Science and Brain-inspired Technology
{ymlei, jpzhang, hmshan}@fudan.edu.cn,
zilongli23@m.fudan.edu.cn, yyli@amss.ac.cn

## Abstract

Interpreting the decisions of deep learning models has been actively studied since the explosion of deep neural networks. One of the most convincing interpretation approaches is salience-based visual interpretation, such as Grad-CAM, where the generation of attention maps depends merely on categorical labels. Although existing interpretation methods can provide explainable decision clues, they often yield partial correspondence between image and saliency maps due to the limited discriminative information from one-hot labels. This paper develops a Language-Image COnsistency model for explainable image classification, termed LICO, by correlating learnable linguistic prompts with corresponding visual features in a coarse-to-fine manner. Specifically, we first establish a coarse global manifold structure alignment by minimizing the distance between the distributions of image and language features. We then achieve fine-grained saliency maps by applying optimal transport (OT) theory to assign local feature maps with class-specific prompts. Extensive experimental results on eight benchmark datasets demonstrate that the proposed LICO achieves a significant improvement in generating more explainable attention maps in conjunction with existing interpretation methods such as Grad-CAM. Remarkably, LICO improves the classification performance of existing models without introducing any computational overhead during inference. Source code is made available at https://github.com/ymLeiFDU/LICO.

## 1 Introduction

Although deep neural networks (DNNs) have shown excellent performance in many fields, the lack of interpretability is still a barrier to landing in some high-stakes scenarios such as medical diagnosis, autonomous driving, *etc*. Therefore, the literature proposes various interpretation methods for DNNs and reveals the decision clues of DNNs to some extent.

Popular interpretation methods can be roughly categorized into two types: 1) gradient back-propagation-based, and 2) class activation mapping (CAM)-based. Both of them mainly take image classification as the pretext task and then generate explainable noisy gradients and saliency maps, respectively. As illustrated in Fig. 1(a), CAM-based methods often explore a better weighting scheme for integrating feature maps of a given input image. The gradient-based methods, in Fig. 1(b), also

---

*Corresponding author.

37th Conference on Neural Information Processing Systems (NeurIPS 2023).

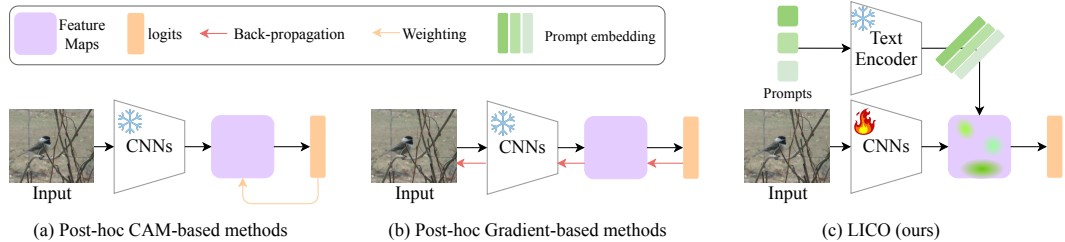

(a) Post-hoc CAM-based methods    (b) Post-hoc Gradient-based methods    (c) LICO (ours)

Figure 1: Motivation of LICO. (a) Existing post-hoc CAM-based methods focus on generating better weighting schemes to obtain weighted-sum attention maps. (b) Gradient-based methods tend to back-propagate gradients from logits to input space. (c) Proposed LICO incorporates learnable prompts to enable image features to approximate semantic information in latent space.

start from the output logits while back-propagating gradients to the input space. However, they are all post-hoc approaches that investigate a DNN pre-trained on a specific dataset [1, 2, 3, 4, 5, 6, 7], and yield biased interpretations due to the limited semantic information that a set of one-hot labels can offer. In addition, one-hot labels often trigger overfitting of the pre-trained model so that the effectiveness of existing interpretation methods could be compromised. On the other hand, the latent feature space of such pre-trained models is unable to sense real semantic space reflecting crucial parts in images. Therefore, it may be futile to explore complex post-hoc techniques for interpreting a pre-trained model.

Inspired by advanced large vision-language models (VLMs) such as CLIP [8], which developed generalized image and text encoders through training on huge amounts of image-text pairs, in this paper, we assume that the large VLMs can encode real-world semantic knowledge through contrastive vision-language alignments, whereas the traditional models pre-trained on relatively small datasets, such as ImageNet-1k, are inferior in capturing the true semantic information.

**Motivation.**    The DNN-based image classification framework generally consists of a convolutional neural network (CNN) for feature extraction and a linear layer acting as a classifier. Since feature representations are used as the input to the classifier, if the DNN is truncated from feature representations, it becomes a linear classifier that aims to classify feature representations linearly into discrete one-hot label space. More specifically, the feature representation lies in the learned manifolds of high-dimensional semantic space [9]. Unfortunately, training using cross-entropy loss with one-hot labels cannot guarantee that the manifolds of image features can reflect the distribution of real-world semantic knowledge, which hinders performance improvement of existing interpretation methods.

In this paper, we leverage language information from large VLMs to enhance current interpretation methods for achieving more explainable saliency maps while enabling promising classification performance improvements; see Fig. 1(c). First, we propose Language-Image-COnsistent (LICO) learning to facilitate the alignment between the manifolds of visual features and class-aware language information. To address the discrete nature of categorical classes, we construct a learnable prompt for each class and map all prompts into a continuous space using the CLIP text encoder, which is feasible to align manifolds of both image and text. Second, we impose each prompt token to correlate with certain feature maps. Considering that the feature maps are redundant to the final classification decision, we propose to encourage the context tokens to guide certain feature maps through distribution alignment using optimal transport (OT) theory.

**Contributions.**    We summarize the main contributions of this paper as follows. (**i**) We propose a novel framework to enhance current interpretation methods by introducing language guidance from large VLMs. (**ii**) We model the discrete categorical class to a continuous space by constructing class-aware learnable prompts, hence, enabling consistent manifold matching between image features and text features. (**iii**) To ensure consistent local feature alignment, we utilize OT theory to reduce the distance between distributions of image and text. (**iv**) Extensive experimental results on eight classification datasets demonstrate the superiority of the proposed LICO against current interpretation methods in terms of quantitative and qualitative results. For practical applications, LICO does not introduce any computational overhead during inference while maintaining improved performance.

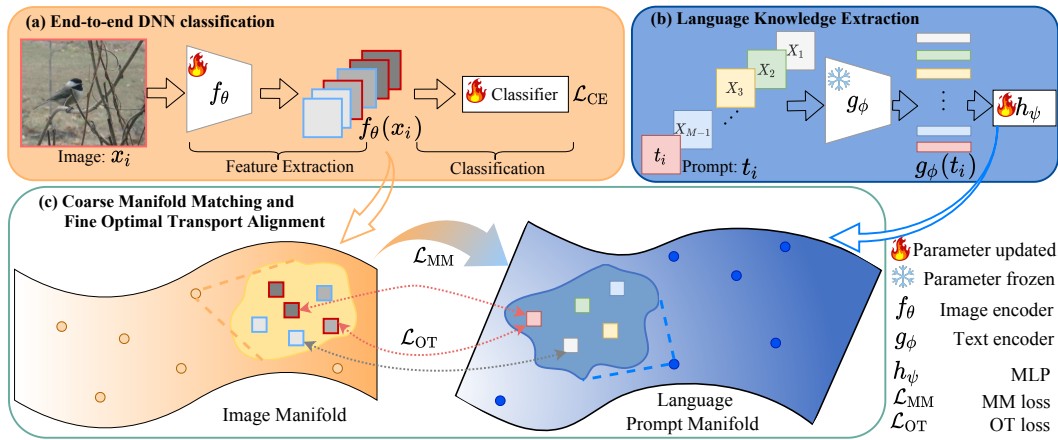

Figure 2: Framework of the proposed LICO. (a) Conventional classification pipeline of DNNs. (b) Language feature extraction with pre-trained text encoder. (c) Manifold matching among samples and optimal transport alignment between feature maps and prompt tokens within each sample.

## 2 Related Work

**Interpretation methods.** Class activation mapping (CAM) is a simple approach to generating class-aware saliency maps for CNNs [1]. Inspired by CAM's effectiveness, its variants including Grad-CAM [2], Grad-CAM++ [3], Score-CAM [4], RISE [10], and Group-CAM [5], were proposed to enhance the precision and interpretability of CNNs. Gradient-based methods such as Guided Back-propatation [11], SmoothGrad [7], and Integrated Gradient (IG) [6] compute the gradients from output logits back to input space. However, they are prone to yielding attribution noise that is inconsistent with human intuition. Some advanced techniques have been developed to reduce attribution noise by utilizing improved path integration theory, such as adaptive path method [12] and important direction method [13]. Distinct from existing interpretation methods that focused on the post-processing of feature map weighting scheme and accurate gradients calculation based on a pre-trained model, our LICO aims to incorporate class-aware learnable prompts that reflect real-world semantic knowledge to guide latent image features. It is worth noting that LICO is compatible with current methods and can consistently improve their performance.

**Prompt learning.** Prompt learning stems from natural language processing (NLP), enabling large language models to directly model the prediction probability of raw text [14, 15, 16]. Following the proliferation of large VLMs, prompt learning began to surface in the realm of computer vision, significantly enhancing multi-modal tasks and text-guided image generation. To overcome the limitations of learning with fixed prompts used in CLIP, learnable prompts have been explored to provide more generalized performance on downstream tasks. CoOp firstly proposed a framework that constructs learnable prompts to align image and text in latent space [17]. Furthermore, to enable CoOp with a stronger capability to perceive the knowledge of new classes, the authors proposed CoCoOp in which the prompts are learned conditioned on image features [18]. PLOT correlated one given image with multiple prompts of its class by optimal transport, leading different prompts to reflect features with respect to different regions within this image [19]. In our LICO, we propose that learnable prompt tokens in one sentence should be registered to certain parts of an image, as shown in Fig. 1(c).

**Optimal transport.** Optimal transport (OT) is a typical metric for measuring discrepancy between two distributions, requiring solving a linear programming problem [20, 21]. Thanks to some fast alternatives, OT has been widely utilized in deep learning, including graph cross-domain alignment [22, 23], domain adaptation [24, 25, 26], optimal graph matching for vessel image registration [27], and multi-modal distribution alignment [28, 29]. Although KL-divergence can effectively measure the cost of which the predicted distribution approximates ground-truth distribution, given the normalized image and text distributions, it is intractable to guarantee they share a common metric space. Therefore, we propose to utilize OT to tackle fine-grained cross-modal alignment.

# 3 Methodology

## 3.1 Overview of LICO

Fig. 2 presents the overview of LICO framework. Given a mini-batch of $B$ training samples $\{(\boldsymbol{x}_i, y_i, t_i)\}_{i=1}^{B}$, the input image $\boldsymbol{x}_i$ is of size $H \times W \times C$, where $C$ is the number of channels, $H$ and $W$ are height and width, respectively. $y_i \in \mathcal{C}$ denotes the class label, and $t_i$ represents the text of corresponding label $y_i$. We denote the pre-trained CLIP text encoder as $g_{\phi}$ while we fixed its parameters $\phi$ during training. The image encoder is denoted as $f_{\theta}$ that is to be optimized. The image encoder takes $\boldsymbol{x}_i$ as input and outputs feature maps $\boldsymbol{F}_i$ with $N$ channels. Note that $\boldsymbol{F}_i$ is a tensor, which is then flattened to one-dimension for each channel; *i.e.*, $\boldsymbol{F}_i \in \mathbb{R}^{N \times d'}$. For the language modeling branch, the text encoder $g_{\phi}$ takes as input the constructed learnable prompt: $\boldsymbol{t}_i = [X_1, X_2, \ldots, X_{M-1}, t_i]$, where $[\ldots]$ denotes the concatenation, $t_i$ is the text corresponding to label of the $i$-th sample, $X_m$ are learnable context tokens, and $M-1$ is the number of context tokens. Through the pre-trained text encoder, the prompt vector $\boldsymbol{t}_i$ is mapped into a continuous $d$-dimension space; *i.e.*, $d = 512$ in CLIP.

In order to measure the distances between image and language features in latent space, we append a mapping net $h_{\psi}$, a multilayer perceptron (MLP) with only one hidden layer, to map the $d$-dimension language features $\boldsymbol{t}_i \in \mathbb{R}^{M \times d}$ to the space of $d'$-dimension, *i.e.*, $\boldsymbol{G}_i \in \mathbb{R}^{M \times d'}$, which is the same as that of image features $\boldsymbol{F}_i$. The structure of $h_{\psi}$ varies along different image encoders. We use $h_{\psi}[a, b]$ to describe its structure, *i.e.*, numbers of hidden units and output units are $a$ and $b$, respectively. Note that in this paper, we do not intend to modify the structure of the existing image and text encoders so that we can obtain the feature maps and corresponding attention maps that are comparable to existing methods. Consequently, we utilize $\boldsymbol{F}_i$ and $\boldsymbol{G}_i$ to preserve the language-image-consistent structure in latent space using manifold matching (Section 3.2) and fine-grained feature-language alignment by optimal transport (Section 3.3).

## 3.2 Language-Image Manifold Matching

To preserve the global manifold structure in latent space, we first measure the relationships, *i.e.* the distances or similarities, among training images and corresponding class-aware prompts.

Although the class categories are discretely distributed that it is infeasible to capture their manifold, we can map them into a continuous space by constructing a prompt vector for each class, then mapping it to a latent space using a pre-trained text encoder. Therefore, we can enforce the image manifold [30] to approximate the language manifold.

In practice, we align the adjacent matrices of language and image features. Specifically, the language adjacent matrix is denoted as $\boldsymbol{A}_{B \times B}^{G}$, and that of image is $\boldsymbol{A}_{B \times B}^{F}$:

$$\boldsymbol{A}_{i,j}^{F} = \frac{\exp(-D(\boldsymbol{F}_i, \boldsymbol{F}_j)/\tau)}{\sum_{s=1}^{B} \exp(-D(\boldsymbol{F}_i, \boldsymbol{F}_s)/\tau)}, \qquad \boldsymbol{A}_{i,j}^{G} = \frac{\exp(-D(\boldsymbol{G}_i, \boldsymbol{G}_j)/\tau)}{\sum_{s=1}^{B} \exp(-D(\boldsymbol{G}_i, \boldsymbol{G}_s)/\tau)}, \qquad (1)$$

where $D(\cdot, \cdot)$ calculates the distance between two images or two prompts, such as Euclidean distance, $\tau$ is the temperature to be learned during training. Then, each image is formulated as a distribution $\boldsymbol{A}_{i,:}^{F}$ where each dimension denotes the distance from the $i$-th sample to others within a mini-batch. Similar to the class-wise prompts, $\boldsymbol{A}_{i,:}^{G}$ implies the relationships between the class prompt of the $i$-th sample and those of others.

Recall our assumption that the large amounts of image-text pairs used in large VLMs, such as CLIP, can lead to a generalized text encoder that well establishes a real-world semantic space. Therefore, for certain downstream tasks and datasets, we aim to introduce the knowledge of this semantic space to the latent space of the target image domain. Then, we propose a manifold matching loss $\mathcal{L}_{\text{MM}}$, which enables the manifold of image features to approach that of prompt features using KL-divergence:

$$\mathcal{L}_{\text{MM}} = \frac{1}{B} \sum_{i=1}^{B} \text{KL}[\boldsymbol{A}_{i,:}^{G} \| \boldsymbol{A}_{i,:}^{F}]. \qquad (2)$$

We note that we do not consider the inverse version $\text{KL}[\boldsymbol{A}_{i,:}^{F} \| \boldsymbol{A}_{i,:}^{G}]$ since LICO focuses on enabling image manifold to approach the manifold of language prompts.

## 3.3 Feature Distribution Alignment by Optimal Transport

While we strive for a coarse alignment of the global manifold structure, it is crucial to establish a correlation between prompt tokens and specific feature maps for each sample, which aids in mitigating the influence of redundant features on the generation of attention maps. Most importantly, the critical feature maps that are most related to the target class should possess the highest similarity with respect to the class token.

Unfortunately, it is challenging to determine or assign which of the $N$ feature maps are correlated with certain prompt tokens. In this paper, we propose to align $\boldsymbol{G}_i = [\boldsymbol{g}_1, \boldsymbol{g}_2, \ldots, \boldsymbol{g}_{M-1}, \boldsymbol{g}_{t_i}]^\top \in \mathbb{R}^{M \times d'}$ and $\boldsymbol{F}_i = [\boldsymbol{f}_1, \boldsymbol{f}_2, \ldots, \boldsymbol{f}_N]^\top \in \mathbb{R}^{N \times d'}$ for achieving consistency between feature maps and specific prompt tokens. In other words, different words in a sentence should correspond to parts of an image, and in contrast, partial regions in an image reflect the semantic information delivered by certain tokens. Therefore, it is intractable to measure this distance using KL-divergence since it is not a strict metric, *i.e.*, it does not satisfy the property of triangle inequality. In LICO, we use optimal transport, which is widely used in measuring distances of two distributions, to align distributions of normalized visual features and prompt features.

For the given image feature maps $\boldsymbol{F}_i \in \mathbb{R}^{N \times d'}$ and a prompt tokens $\boldsymbol{G}_i \in \mathbb{R}^{M \times d'}$, we construct two discrete distributions:

$$\boldsymbol{\mu} = \sum_{n=1}^{N} u_n \delta_{\boldsymbol{f}_n}, \qquad \boldsymbol{v} = \sum_{m=1}^{M} v_m \delta_{\boldsymbol{g}_m}, \tag{3}$$

where $\delta_{\boldsymbol{f}_n}$ is a Dirac function centered at $\boldsymbol{f}_n$, so as to $\delta_{\boldsymbol{g}_m}$, and the weights $\boldsymbol{u} = \{u_n\}_{n=1}^{N} \in \Delta_N$ and $\boldsymbol{v} = \{v_m\}_{m=1}^{M} \in \Delta_M$, $\Delta$ denotes the $N$- and $M$-dimensional probability simplex, *i.e.*, $\sum_{n=1}^{N} u_n = 1$, $\sum_{m=1}^{M} v_m = 1$. Finally, the discrete OT distance for one sample is defined as follows:

$$D_{\text{OT}}(\boldsymbol{\mu}, \boldsymbol{v}) = \inf_{\boldsymbol{\pi} \in \Pi(\boldsymbol{\mu}, \boldsymbol{v})} \mathbb{E}_{(\boldsymbol{\mu}, \boldsymbol{v}) \sim \boldsymbol{\pi}}[\boldsymbol{C}(\boldsymbol{f}, \boldsymbol{g})] = \min_{\boldsymbol{T} \in \Pi(\boldsymbol{\mu}, \boldsymbol{v})} \sum_{n=1}^{N} \sum_{m=1}^{M} \boldsymbol{T}_{n,m} \cdot c(\boldsymbol{f}_n, \boldsymbol{g}_m), \tag{4}$$

$$s.t. \quad \boldsymbol{T}\mathbf{1}_m = \boldsymbol{\mu}, \quad \boldsymbol{T}\mathbf{1}_n = \boldsymbol{v},$$

where $\boldsymbol{C} \in \mathbb{R}^{N \times M}$ represents the cost matrix in which each element $c(\boldsymbol{f}_n, \boldsymbol{g}_m)$ denotes transportation cost between $\boldsymbol{f}_n$ and $\boldsymbol{g}_m$. $\boldsymbol{T} \in \mathbb{R}^{N \times M}$ is the transport plan that is to be optimized and $\Pi(\boldsymbol{\mu}, \boldsymbol{v})$ denotes the transportation polytope that contains all joint probabilities of $\boldsymbol{\mu}$ and $\boldsymbol{v}$. In practice, solving the optimization problem in Eq. (4) often equips with a high computation cost. Thus we use the Sinkhorn algorithm, which is more computationally amenable, to solve an entropy-constrained problem [31]:

$$D_{\text{OT}}(\boldsymbol{\mu}, \boldsymbol{v}) = \min_{\boldsymbol{T} \in \Pi(\boldsymbol{\mu}, \boldsymbol{v})} \sum_{n=1}^{N} \sum_{m=1}^{M} \boldsymbol{T}_{n,m} \cdot c(\boldsymbol{f}_n, \boldsymbol{g}_m) - \lambda \mathbb{H}(\boldsymbol{T}), \quad s.t. \quad \boldsymbol{T}\mathbf{1}_m = \boldsymbol{\mu}, \quad \boldsymbol{T}\mathbf{1}_n = \boldsymbol{v}, \tag{5}$$

where $\lambda$ is Lagrange multiplier and $\mathbb{H}(\boldsymbol{T}) = \sum_{n,m} \boldsymbol{T}_{n,m} \log \boldsymbol{T}_{n,m}$. Then after a few iterations, we obtain the optimal solutions:

$$\boldsymbol{T}^* = \text{diag}(\boldsymbol{\mu}^t) \exp(-\boldsymbol{C}/\lambda) \text{diag}(\boldsymbol{v}^t), \tag{6}$$

where $t$ is the iteration step. $\boldsymbol{\mu}^t$ and $\boldsymbol{v}^t$ are updated according to following rules:

$$\boldsymbol{\mu}^t = \boldsymbol{\mu}/(\exp(-\boldsymbol{C}/\lambda)\boldsymbol{v}^{t-1}), \quad \boldsymbol{v}^t = \boldsymbol{v}/(\exp(-\boldsymbol{C}/\lambda)^\top \boldsymbol{\mu}^t). \tag{7}$$

**Dynamic context (DC).** To endow each image with diverse prompt tokens, we shuffle the learnable context tokens in each training iteration referring to the training procedure in Algorithm 1.

### 3.4 Final Objective Function

The final training loss function is as follows:

$$\mathcal{L} = \mathcal{L}_{\text{CE}} + \alpha \mathcal{L}_{\text{MM}} + \beta \mathcal{L}_{\text{OT}}, \tag{8}$$

where $\mathcal{L}_{\text{CE}}$ is the cross-entropy loss, $\mathcal{L}_{\text{OT}} = \frac{1}{B} \sum_{i=1}^{B} D_{\text{OT}}$, $\alpha$ and $\beta$ are hyperparameters for adjusting different terms. During the inference phase, we apply the trained image encoder and classifier to conduct conventional classification that yields the predicted probability of a given input image. Note that the text encoder and the MLP mapping do not affect the inference procedure. The detailed algorithm can be found in Algorithm 1.

**Algorithm 1** Training Algorithm of LICO.

---

**Require:** Training set $\mathcal{S}$, total epochs $U$, image encoder $f_{\boldsymbol{\theta}}$, text encoder $g_{\boldsymbol{\phi}}$, MLP $h_{\boldsymbol{\psi}}$, learnable prompts $\boldsymbol{t}_i = [X_1, X_2, \ldots, \ldots, X_{M-1}, t_i]$.

**Return:** Image encoder with optimal parameter $\boldsymbol{\theta}^*$.

1: **for** $u = 1$ to $U$ **do**
2:     Sample a mini-batch of $(\{\boldsymbol{x}_i, \boldsymbol{t}_i, y_i\}_{i=1}^B)$ from $\mathcal{S}$.
3:     Randomly shuffling $X_m$ and $t_i$.                        $\triangleright$ Dynamic context
4:     $\boldsymbol{F}_i = f_{\boldsymbol{\theta}}(\boldsymbol{x}_i), \boldsymbol{G}_i = h_{\boldsymbol{\psi}}(g_{\boldsymbol{\phi}}(\boldsymbol{t}_i))$.         $\triangleright$ Image and text features
5:     Calculate $\boldsymbol{A}_{B \times B}^G$, $\boldsymbol{A}_{B \times B}^F$                    $\triangleright$ Adjacent matrices
6:     Calculate $\mathcal{L}_{\text{MM}}$ according to Eq. (2).     $\triangleright$ Coarse alignment by manifold
7:     Optimal transport plan $\boldsymbol{T}^*$ by Eq. (6), then calculate $\mathcal{L}_{\text{OT}}$. $\triangleright$ Fine-grained alignment by OT
8:     Classifier $\longleftarrow \boldsymbol{F}_i$
9:     Total loss: $\mathcal{L} = \mathcal{L}_{\text{CE}} + \alpha \mathcal{L}_{\text{MM}} + \beta \mathcal{L}_{\text{OT}}$, update $\boldsymbol{\theta}, \boldsymbol{\psi}$, and $X_m$ by gradients: $\frac{\partial \mathcal{L}}{\partial \boldsymbol{\theta}}, \frac{\partial \mathcal{L}}{\partial \boldsymbol{\psi}}, \frac{\partial \mathcal{L}}{\partial \boldsymbol{X}}$.
10: **end for**

---

## 4 Experiments

### 4.1 Datasets

This paper focuses on image classification task and evaluates the proposed LICO on well-known datasets, including ImageNet-1k [32], CIFAR-10/100 [33], and SVHN [34]. We conduct the classification experiments under the setting of limited training data in which the splits of labeled data follow the previous works for fair comparison [35, 36]. Furthermore, we conduct fine-grained classification on typical benchmarks, including CUB-200 [37], FGVC-Aircraft [38], Stanford Cars-196 [39], VGG Flowers [40]. The evaluation is carried out on both the full dataset and few-shot settings, following the procedure of CGC [41].

### 4.2 Implementation Details

We employ the ViT-B/32 trained by CLIP [8] as the text encoder and its parameters are fixed during training. The output dimension of this text encoder is 512 for each token. The image encoders in our experiments vary along different datasets and training settings. Specifically, for the ImageNet experiments, we utilize ResNet-50 as the image classifier [42]. In doing so, the convolutional layers preceding the final linear layer constitute the image encoder of LICO. For the CIFAR-10/100 and SVHN, we follow the experimental settings in [35, 36], the classification network is Wide ResNet (WRN) [43]. We further conduct the same experiments using another network, the PreAct-ResNet-18 (PARN-18) [44]. For fine-grained classifications on CUB-200, Standford-Cars, Aircraft, and VGG Flowers datasets, we applied the same settings used in CGC for fair comparison [41], where the image encoder is also the ResNet-50 that has been pre-trained on ImageNet-1k with CE loss. Note that for all the experiments of LICO, we only use the trained image encoder and the classifier during the inference phase. The text encoder, learnable prompt contexts, and MLP mapping net are dropped, thus, will not compromise the computational efficiency. Hyperparameters $\alpha$ and $\beta$ are set as 10 and 1, respectively.

All the experiments are implemented by PyTorch [45]. The learning rates for ImageNet, CIFAR-10/100, and SVHN are of 0.03 with a consine rate decay schedule, *i.e.*, $\eta = \eta_0 \cos(\frac{7\pi k}{16K})$, where $\eta_0$ denotes the initial learning rate and $k$ is the index of training step [46]. We use a standard stochastic gradient descent (SGD) optimizer with a momentum of 0.9 [47, 48], and the weight decay is 0.0001. The training batch sizes are 128 and 64 for ImageNet and other datasets, respectively. Specifically, the mapping net for ResNet-50 is $h_{\boldsymbol{\psi}}[512, 49]$, $h_{\boldsymbol{\psi}}[512, 64]$ for WRN, and $h_{\boldsymbol{\psi}}[512, 49]$ for PARN-18. The total training epoch is 90 for ImageNet and 200 for others. The experiments were trained on four NVIDIA A100 GPUs for ImageNet-1k and one GPU for other datasets.

### 4.3 Comparison of Interpretation Capability

In this experiment, we compare interpretation results of popular methods, including Grad-CAM [2], Grad-CAM++ [3], RISE [10], Score-CAM [4], Group-CAM [5], and CGC [41]. Note that LICO is compatible with these post-hoc interpretation methods so that in our experiments, we compare the

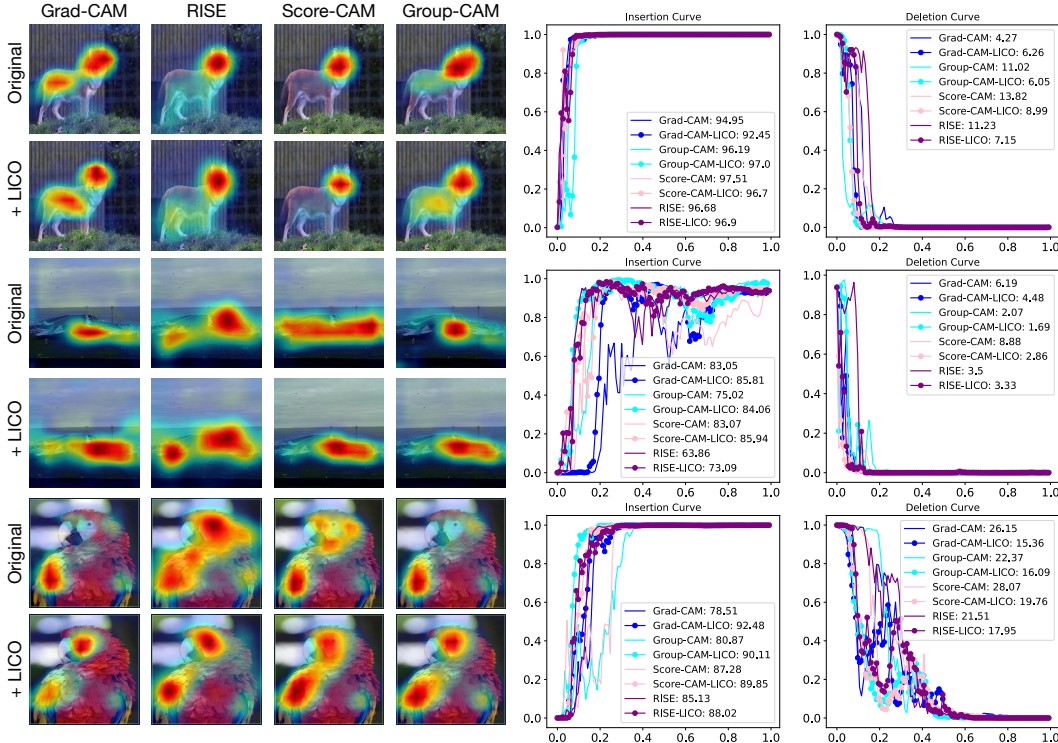

Figure 3: Saliency maps (Left) and Insertion/Deletion curves (Right) on ImageNet-1k validation set.

saliency maps obtained by these interpretation methods using baseline ImageNet pre-trained model and LICO trained counterparts of the same architectures.

**Qualitative results.** Qualitatively, we compare the saliency maps obtained by different interpretation methods in Fig. 3 (Left). We can see that for single object images, LICO can effectively help baseline methods cover more comprehensive and discriminative regions, *e.g.*, the head and fur of a bird. However, the baseline methods are inferior in capturing all the objects in the multi-object image. Please refer to Appendix A for more results of multi-class and multi-object cases.

Table 1: Quantitative comparisons of different interpretation methods on ImageNet in terms of Insertion and Deletion. We report the results: "Baseline/+LICO". $Overall = Insertion - Deletion$.

| Method | Grad-CAM++ | Grad-CAM | RISE | Score-CAM | Group-CAM | CGC |
|---|---|---|---|---|---|---|
| Insertion↑ | 50.0/**51.2** | 53.5/**57.1** | 54.0/**54.9** | 55.1/**55.6** | **56.8**/55.2 | 52.2/**55.4** |
| Deletion↓ | 14.8/**11.7** | **13.3**/15.1 | 11.7/**10.8** | 11.5/**11.2** | 12.3/**10.5** | -/15.8 |
| Overall↑ | 35.2/**39.5** | 40.2/**42.0** | 43.6/**44.1** | 42.3/**44.4** | 44.5/**44.7** | -/39.6 |

**Quantitative results.** To achieve quantitative evaluation, we follow [10, 4, 5] to conduct Insertion and Deletion tests. Insertion gradually introduces class-related regions (3.6% pixels) of an original image to a blurred image according to the values of the saliency map. This process is repeated until the blurred image is fully recovered. In contrast, Deletion aims to replace related pixels (3.6%) in a blurred image with those of the corresponding original image. In Table 1, we provide the AUC of the classification score after Softmax. For most of the interpretation methods, LICO consistently improves the Insertion and Deletion values. Although the insertion value of Group-CAM and deletion value of Grad-CAM is better than LICO, the LICO still achieves the best overall values. We also report the quantitative results of corresponding cases with different interpretation methods in Fig. 3 (Right). Please refer to Appendix B for the experiment of Pointing Game [5].

**Sanity checks.** Sanity check for saliency maps was first proposed in [49], which is a qualitative test aiming at evaluating whether the saliency maps are sensitive to model parameters. We conducted two types of test: cascading randomization from top to bottom layers and independent randomizing of

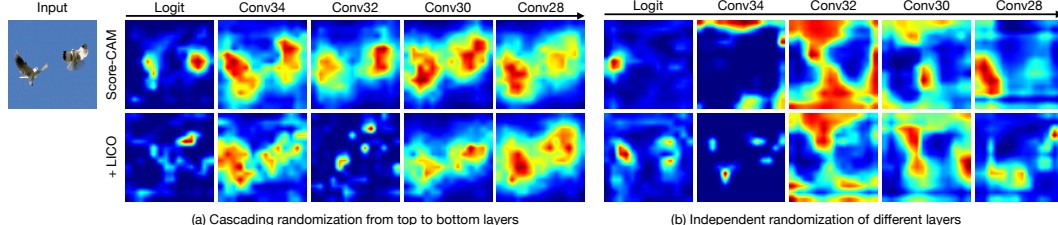

| Input | Logit | Conv34 | Conv32 | Conv30 | Conv28 | Logit | Conv34 | Conv32 | Conv30 | Conv28 |

Score-CAM / +LICO

(a) Cascading randomization from top to bottom layers     (b) Independent randomization of different layers

Figure 4: Sanity checks evaluated using ImageNet pre-trained VGG-19 and LICO trained version. Each column represents saliency maps after randomizing corresponding layers following two randomization schemes: (a) cascading randomization from top to bottom layers, and (b) independent randomization of different layers.

different layers. In this experiment, Score-CAM is selected as the baseline method, and the backbone network is VGG-19. In Fig. 4, we can see that Score-CAM is sensitive to the model parameters, and the saliency maps obtained by LICO-trained model also pass these two tests. From Fig. 4(a) we can see that the saliency maps of LICO are more sensitive to model parameters along cascaded randomization, *e.g.*, from `Logit` to `Conv34`, then to `Conv32`.

### 4.4 Classification Performance

In addition to interpretation comparison, we compare the classification performance on various datasets since LICO can act as a regularization for enhancing image classifier. According to previous studies [1, 41, 50], a good interpretation method often sacrifices the discriminative ability, *i.e.*, lower classification accuracy than baselines. As shown in Table 2, we compare non-post-hoc methods GCC [50] and CGC [41] with our LICO. Despite their emphasis on interpretation ability, both GCC and CGC experience performance degradation compared to the CE loss pre-trained model. In contrast, *LICO surpasses the baseline model and achieves the highest Top-1 and Top-5 accuracy values, a notable advantage not shared by previous interpretation methods*. Although LICO introduces relatively few amounts of training parameters, *i.e.*, MLP $h_\psi$ and prompt tokens $X_m$, which is trivial to the enhancement of both interpretation and classification performance. Furthermore, it does not affect model architectures and introduces any computational overhead at the inference phase. Please refer to Appendices C and D for further evaluation of LICO using ViTs and the potential in regularizing semantic space obtained by $t$-SNE, respectively.

Table 2: Classification accuracy (%) on ImageNet using ResNet.

| Method | Top-1 | Top-5 |
|---|---|---|
| ResNet-50 | 76.13 | 92.91 |
| +GCC [50] | 74.40 | 92.12 |
| +CGC [41] | 74.60 | 92.24 |
| **+LICO** | **76.27** | **92.99** |
| ResNet-18 | 69.76 | 89.08 |
| +GCC [50] | 67.74 | 88.52 |
| +CGC [41] | 66.37 | 88.27 |
| **+LICO** | **69.83** | **89.21** |

**Training with limited data.** In Table 3, we evaluate the performance of LICO with limited training data. Following [36, 35], we conduct experiments on CIFAR-10/100 and SVHN with different label amounts. Note that we only use limited labeled data for training and do not focus on whether LICO is useful under a semi-supervised setting. Across all the limited training settings, LICO consistently improves both PARN18 and WRN baselines. Notably, in the full training setting, LICO exhibits the largest improvement on CIFAR-100, which is attributed to the effectiveness of prompt guidance introduced by manifold of larger amounts of classes.

Table 3: Classification accuracies (%) on CIFAR-10/100 and SVHN datasets using different amounts of labels. "Full" means training with all labeled data.

| Dataset | CIFAR-10 | | | CIFAR-100 | | | SVHN | |
|---|---|---|---|---|---|---|---|---|
| Labels | 40 | 250 | 4000 | 400 | 2500 | 10000 | 40 | 1000 |
| PARN18 | $22.9_{\pm0.23}$ | $32.9_{\pm0.42}$ | $77.5_{\pm0.63}$ | $5.2_{\pm0.07}$ | $23.7_{\pm0.17}$ | $54.9_{\pm0.14}$ | $14.7_{\pm0.27}$ | $56.2_{\pm0.21}$ |
| **+LICO** | $\mathbf{23.7_{\pm0.12}}$ | $\mathbf{36.1_{\pm0.61}}$ | $\mathbf{80.1_{\pm0.05}}$ | $\mathbf{5.8_{\pm0.14}}$ | $21.4_{\pm0.48}$ | $\mathbf{56.5_{\pm0.67}}$ | $\mathbf{16.0_{\pm0.19}}$ | $\mathbf{59.5_{\pm0.41}}$ |
| WRN | $23.4_{\pm0.92}$ | $36.9_{\pm1.08}$ | $80.8_{\pm0.07}$ | $8.6_{\pm0.17}$ | $32.3_{\pm0.66}$ | $60.7_{\pm0.24}$ | $15.9_{\pm0.23}$ | $58.9_{\pm0.22}$ |
| **+LICO** | $\mathbf{24.9_{\pm0.42}}$ | $\mathbf{41.7_{\pm0.91}}$ | $\mathbf{81.5_{\pm0.43}}$ | $\mathbf{10.7_{\pm0.51}}$ | $\mathbf{35.5_{\pm0.40}}$ | $\mathbf{62.0_{\pm0.48}}$ | $\mathbf{17.4_{\pm0.34}}$ | $\mathbf{63.7_{\pm0.34}}$ |
| WRN_Full | | $95.6_{\pm0.12}$ | | | $80.9_{\pm0.18}$ | | | $97.9_{\pm0.02}$ |
| **+LICO** | | $\mathbf{95.8_{\pm0.08}}$ | | | $\mathbf{82.2_{\pm0.02}}$ | | | $\mathbf{98.3_{\pm0.06}}$ |

Table 4: Fine-grained classification accuracy under full and few-shot settings.

| Method | 1-shot | 5-shot | 10-shot | Full | 1-shot | 5-shot | 10-shot | Full |
|---|---|---|---|---|---|---|---|---|
| | CUB-200 | | | | Standford-Cars | | | |
| Baseline | $13.7_{\pm0.3}$ | $51.7_{\pm0.3}$ | $66.4_{\pm0.2}$ | $80.1_{\pm0.9}$ | $6.1_{\pm0.2}$ | $34.3_{\pm0.4}$ | $61.1_{\pm0.4}$ | $89.7_{\pm0.1}$ |
| +CGC [41] | $15.8_{\pm0.3}$ | $55.2_{\pm0.3}$ | $\mathbf{68.4}_{\pm0.3}$ | $81.5_{\pm0.1}$ | $6.5_{\pm0.2}$ | $36.5_{\pm0.4}$ | $63.0_{\pm0.4}$ | $90.3_{\pm0.1}$ |
| **+LICO** | $\mathbf{16.9}_{\pm0.2}$ | $\mathbf{55.8}_{\pm0.4}$ | $\mathbf{68.4}_{\pm0.2}$ | $\mathbf{82.7}_{\pm0.3}$ | $\mathbf{7.1}_{\pm0.3}$ | $\mathbf{37.2}_{\pm0.5}$ | $\mathbf{64.4}_{\pm0.4}$ | $\mathbf{91.5}_{\pm0.2}$ |
| | Aircraft | | | | VGG Flowers | | | |
| Baseline | $7.7_{\pm0.3}$ | $25.7_{\pm0.4}$ | $41.4_{\pm0.3}$ | $83.7_{\pm0.2}$ | $52.1_{\pm0.5}$ | $85.6_{\pm0.4}$ | $93.2_{\pm0.2}$ | $96.1_{\pm0.2}$ |
| +CGC [41] | $8.0_{\pm0.3}$ | $26.9_{\pm0.4}$ | $42.9_{\pm0.3}$ | $\mathbf{85.7}_{\pm0.2}$ | $53.3_{\pm0.5}$ | $85.8_{\pm0.4}$ | $\mathbf{93.4}_{\pm0.2}$ | $96.2_{\pm0.2}$ |
| **+LICO** | $\mathbf{8.4}_{\pm0.4}$ | $\mathbf{27.5}_{\pm0.2}$ | $\mathbf{43.1}_{\pm0.4}$ | $85.6_{\pm0.2}$ | $\mathbf{55.6}_{\pm0.4}$ | $\mathbf{86.2}_{\pm0.3}$ | $\mathbf{93.4}_{\pm0.4}$ | $\mathbf{96.8}_{\pm0.3}$ |

**Fine-Grained classification.** Furthermore, we evaluate LICO on a fine-grained classification problem that requires the model to capture fine-grained features. In Table 4, except for the Aircraft, LICO enhances classification performance under all settings compared with CGC. This observation highlights that contrastive learning among attention maps used in CGC primarily emphasizes similarities between target images and other random/augmented images, while disregarding the measurement of semantic distinctions. For the Aircraft dataset, it is difficult for LICO to achieve significant improvements due to the categorical labels are types of aircraft, *e.g.*, the numerical symbols such as 727-200 which are challenging for CLIP text encoder to achieve meaningful embedding, and CLIP has demonstrated relatively lower accuracy on some out of distribution data like aircraft and satellite [8]. To address the challenges, we incorporate prior text knowledge by constructing initial prompts as "a type of aircraft", which allows LICO to improve performance in the few-shot settings while achieving comparable results to CGC under the full setting. This strategy has also been verified in CoCoOp [18].

## 4.5 Ablation Study

**Ablation on manifold matching and OT alignment.** In Table 5, we investigate the effectiveness of $\mathcal{L}_{MM}$ and $\mathcal{L}_{OT}$. We can see that only using $\mathcal{L}_{OT}$ obtains more performance drop than that of $\mathcal{L}_{MM}$.

Table 5: Ablation on $\mathcal{L}_{MM}$ and $\mathcal{L}_{OT}$.

| $\mathcal{L}_{CE}$ | $\mathcal{L}_{MM}$ | $\mathcal{L}_{OT}$ | Top-1 | Top-5 | Insert. | Delet. |
|---|---|---|---|---|---|---|
| ✔ | | | 76.13 | 92.91 | 53.5 | **13.3** |
| ✔ | | ✔ | 75.98 | 92.92 | 56.6 | 16.0 |
| ✔ | ✔ | | 76.18 | 92.90 | 56.9 | 15.5 |
| ✔ | ✔ | ✔ | **76.27** | **92.99** | **57.1** | 15.1 |

This indicates that the global consistency of manifolds guarantees the basic performance, and OT only focuses on local feature alignments so that it cannot be sensitive to relationships between intra- and inter-class samples.

**Ablation on number of context tokens.** In Table 6, we evaluate the performances influenced by different numbers of learnable tokens. We find that 12 is the best choice in our experiments, and the settings of 16 and 20 are relatively better than those of 4 and 8.

**Ablation on distance function** $D$ **in Eq. (1).** In Table 7, we compare the similarity functions used in manifold matching, which measures distances among samples within a mini-batch. We can see that the Euclidean distance is more suitable to our LICO.

Table 6: Ablation on no. of context tokens.

| no. | Top-1 | Top-5 | Insertion | Deletion |
|---|---|---|---|---|
| 0 | 75.64 | 91.92 | 54.1 | 17.8 |
| 4 | 76.03 | 92.74 | 55.2 | 17.5 |
| 8 | 76.09 | 92.89 | 56.3 | 16.0 |
| 12 | **76.27** | **92.99** | **57.1** | **15.1** |
| 16 | 76.21 | 92.87 | 57.0 | 15.8 |
| 20 | 76.14 | 92.93 | 56.9 | 15.5 |

Table 7: Ablation on distance function.

| Dataset | Euclidean | Cosine |
|---|---|---|
| CIFAR-10 | $\mathbf{95.78}_{\pm0.08}$ | $95.46_{\pm0.12}$ |
| CIFAR-100 | $\mathbf{82.22}_{\pm0.02}$ | $81.85_{\pm0.05}$ |
| SVHN | $\mathbf{98.25}_{\pm0.06}$ | $98.21_{\pm0.06}$ |
| CUB-200 | $\mathbf{82.70}_{\pm0.30}$ | $82.10_{\pm0.30}$ |
| Flowers | $\mathbf{96.80}_{\pm0.30}$ | $96.20_{\pm0.40}$ |

Please refer to Appendices E, F, and G for more ablation studies on DC, effects of different type of text encoders, and frozen parameters of prompts, respectively.

## 5 Conclusion

In this paper, we proposed LICO to enhance existing visual interpretation methods by incorporating language information, which is compatible with these methods. The LICO aligns image and prompt embeddings globally using manifold matching, simultaneously aligns feature maps with corresponding learnable context tokens by applying optimal transport alignment. Extensive experiments on evaluating interpretation capability and classification performance exhibit both quantitative and qualitative enhancement introduced by LICO. A key limitation of LICO is that it depends on a trainable MLP to project language embeddings into a metric space of same dimension with that of image features, where the output dimension of such MLPs varies according to different image encoders.

**Broader Impacts** LICO explores effective visual interpretations of DNNs by introducing language knowledge, which is orthogonal to existing post-hoc interpretation methods. An important merit of LICO is to enhance interpretability while achieving competitive or even better classification performance, applicable to various kinds of tasks and models effectively.

## Acknowledgements

This work was supported in part by Shanghai Municipal Science and Technology Major Project (No. 2018SHZDZX01) and ZJLab, Natural Science Foundation of Shanghai (No. 21ZR1403600), National Natural Science Foundation of China (Nos. 62101136, 62306075), China Postdoctoral Science Foundation (No. 2022TQ0069), Young Elite Scientists Sponsorship Program by CAST (No. 2022QNRC001), Shanghai Municipal of Science and Technology Project (No. 20JC1419500), and Shanghai Center for Brain Science and Brain-inspired Technology.

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

# Appendix

## A  More Results of Saliency Maps

**LICO localizes more discriminative local features.**    Fig. 5 provides more saliency maps, showing that LICO is able to localize fine local features of target objects, *e.g.*, the foot and head in Fig. 5(c).

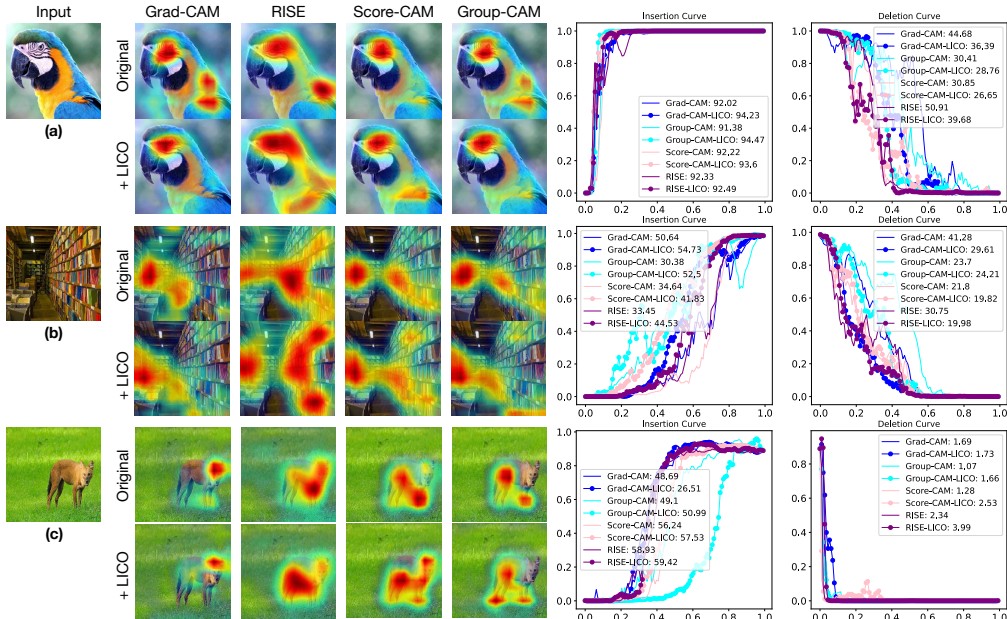

Figure 5: More samples on ImageNet-1k. For each image, we provide saliency maps of different methods and Insertion/Deletion curves. LICO helps localize fine local features.

**More comprehensive locations yield lower metrics.**    Here, we provide some failure cases obtained by our LICO, where the saliency maps of LICO exhibit more comprehensive locations on target objects while resulting in lower AUC values of Insertion and higher AUC values of Deletion compared with baseline methods.

For the cases with multiple target objects in Fig. 6, LICO localizes different objects separately and comprehensively, which demonstrates the consistency between language knowledge and visual features established by our LICO. However, LICO obtains inferior Insertion/Deletion evaluations, which is paradoxical to human cognition. We conjecture that the main reason is attributed to the fixed text encoder, which is not specifically optimized for target tasks, *i.e.*, the images with a single object occupy a higher percentage of the dataset such as ImageNet-1k. Therefore, the located multiple objects may be harmful to saliency maps-guide pixel-level insertion and deletion strategies.

In Fig. 7, we provide some examples that contain multi-class objects in one image, and for the target classes, LICO obtains more accurate localizations. In Fig. 8, we verified the effectiveness of LICO on capturing more objects of sinlge class within one image, and we can see that LICO captures more comprehensive objects than those without LICO.

## B  Pointing Game

To verify the localization ability of LICO, we conducted the pointing game on MS COCO 2017 validation set for localization evaluation. Following settings in Score-CAM and Group-CAM, we quantified localization by calculating $\frac{\text{Hits}}{\text{Hits+Misses}}$ , assessing if salient pixels fall within the annotated bounding boxes. Table 8 shows that LICO consistently improves all the baseline interpretation methods, indicating the effectiveness of regularization by the proposed manifold OT losses.

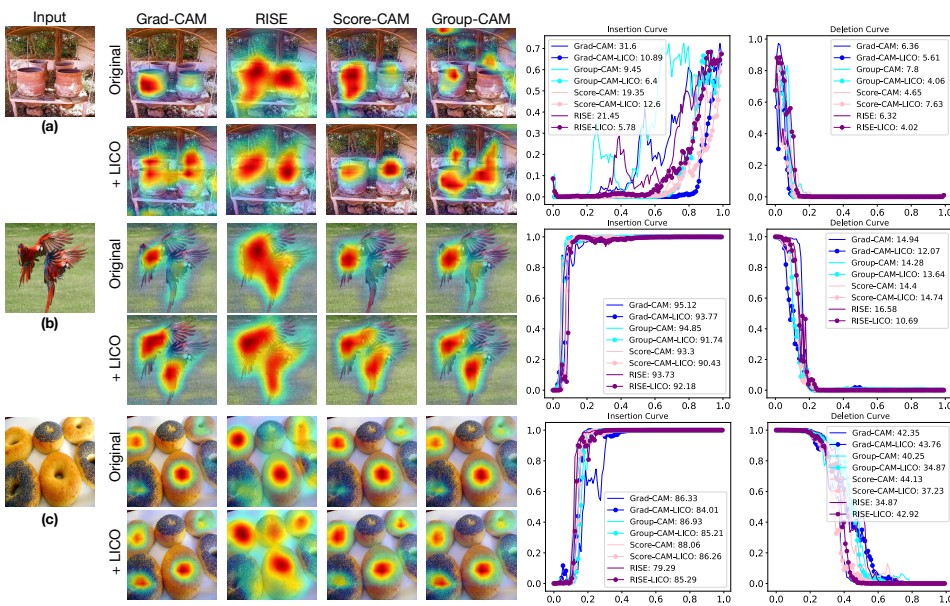

Figure 6: Saliency maps of failure cases with multiple target objects on ImageNet-1k. For each image, we provide saliency maps of different methods and Insertion/Deletion curves.

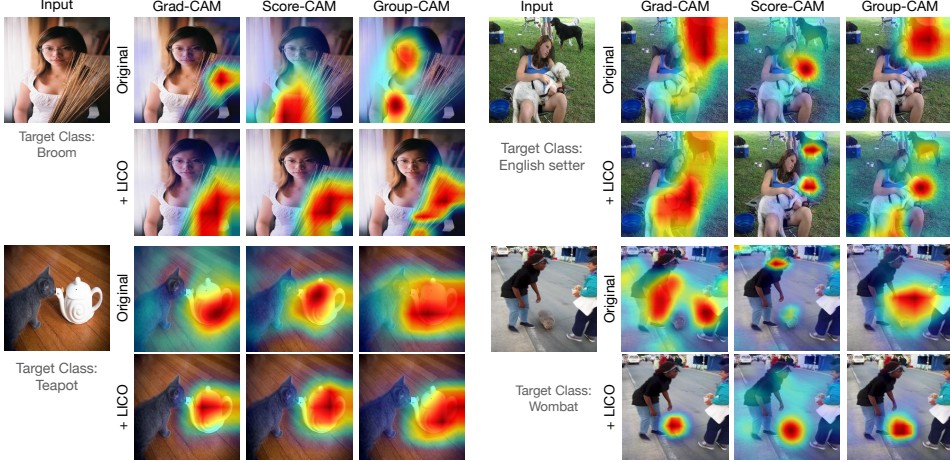

Figure 7: Saliency maps of multi-class cases.

## C Evaluation on ViT

We further trained a ViT-Base-16 network on ImageNet-1k dataset and presented the corresponding accuracy, insertion, and deletion in Table 9. We calculated the $\mathcal{L}_{\mathrm{MM}}$ and $\mathcal{L}_{\mathrm{OT}}$ between language tokens and representations of patch tokens and class tokens. For attention maps, we applied Grad-CAM in LICO-trained ViT-Base-16 by calculating gradients from outputs to the last attention layer of the class token. Table 9 further confirms that the transformer model with LICO not only performs better in classification but also gains better interpretability than the one without LICO, which is in line with the finding for the CNN-based backbone.

## D t-SNE Visualization of Latent Features

We further explore how LICO can regularize the latent space. In Fig. 9, we provide the t-SNE visualization of learned visual features on CIFAR-10 and SVHN. In Fig. 9(a), our LICO enables different classes with similar distribution shapes. Interestingly, if we roughly categorize the ten

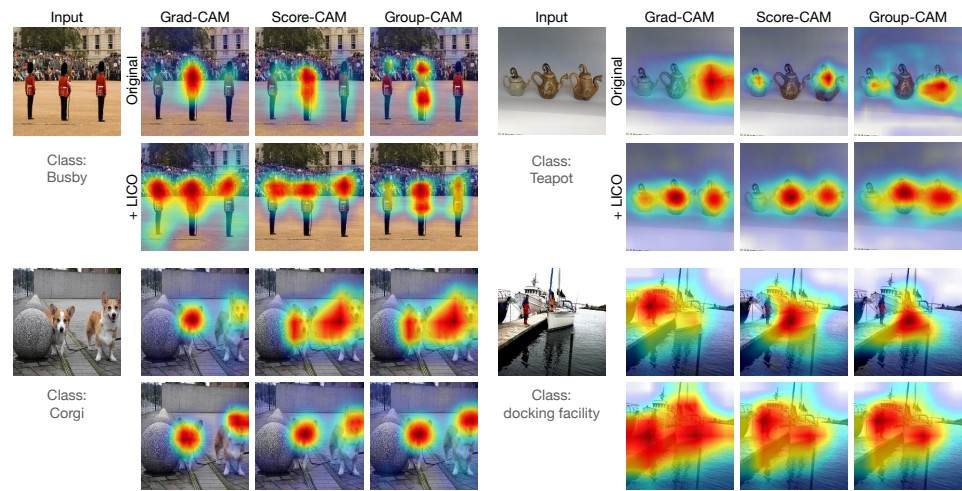

Figure 8: Saliency maps of single-class and multi-object cases.

Table 8: Pointing game evaluation on MS COCO 2017 val set.

| LICO | Grad-CAM | Grad-CAM++ | RISE | XRAI | Score-CAM | Group-CAM |
|---|---|---|---|---|---|---|
| ✗ | $56.7_{\pm 0.225}$ | $57.2_{\pm 0.227}$ | $54.3_{\pm 0.205}$ | $55.1_{\pm 0.232}$ | $51.0_{\pm 0.211}$ | $57.5_{\pm 0.202}$ |
| ✓ | $\mathbf{56.9}_{\pm 0.221}$ | $\mathbf{58.1}_{\pm 0.215}$ | $\mathbf{55.2}_{\pm 0.201}$ | $\mathbf{56.7}_{\pm 0.229}$ | $\mathbf{52.5}_{\pm 0.205}$ | $\mathbf{58.2}_{\pm 0.197}$ |

classes of CIFAR-10 into two classes, *i.e.*, "animal" and "vehicle", our LICO can almost linearly classify these two classes, whereas the baseline obtains non-linear classification boundary (dashed lines). In Fig. 9(b), LICO makes the latent space exhibit an manifold structure with respect to semantically ordinal classes such as from "1" to "4", and some semantically closer classes are also closer in latent space such as "9" and "'10". However, the baseline method is unable to capture the underlying manifold structure and we can see obviously that "7" is far away from "5", "6", and "8", so as to "9" and "10", which should be closer in nature. In Figs. 9(c) and (d), LICO also shows more discriminant results under limited training data.

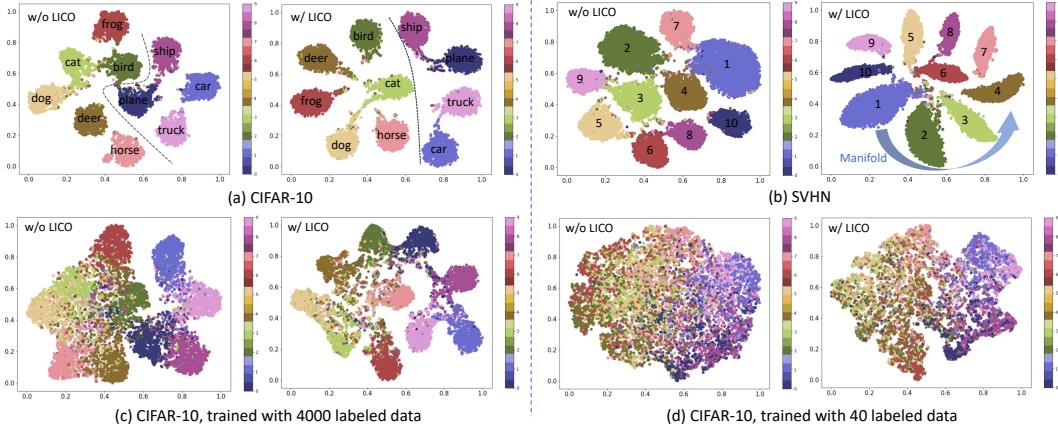

Figure 9: t-SNE results obtained by the models with and without our LICO. (a) and (b) are results of full setting on CIFAR-10 and SVHN, respectively. (c) and (d) show the results under the setting of limited training data.

Table 9: Evaluation on ImageNet-1k using ViT-Base-16.

| Model | Accuracy | Insertion | Deletion |
|---|---|---|---|
| ViT-Base-16 w/o LICO | 77.9 | 55.2 | 14.4 |
| ViT-Base-16 w/ LICO | **78.2** | **56.0** | **13.8** |

Table 10: Ablation on DC.

| DC | ImageNet | CIFAR-10 | CIFAR-100 | SVHN |
|---|---|---|---|---|
| ✗ | 76.22 | $95.45_{\pm0.09}$ | $81.3_{\pm0.05}$ | $98.1_{\pm0.10}$ |
| ✔ | **76.27** | $\mathbf{95.80}_{\pm0.08}$ | $\mathbf{82.2}_{\pm0.02}$ | $\mathbf{98.3}_{\pm0.06}$ |

## E  Ablation Studies

**Ablation on dynamic context (DC).**   In the training algorithm of LICO, we designed a dynamic context (DC) for learnable context. We provide some results on ablation of DC in Tables 10 and 11. We can see that, for most of datasets we evaluated, DC consistently improves the classification performance. For few-shot settings in Table 11, DC obtains slight performance improvements, which is attributed to few samples of each class. That is to say, there are insufficient visual feature variety to endow context vectors with various information.

Table 11: Ablation on DC on four fine-grained datasets.

| | CUB-200 | | | | Standford-Cars | | | |
|---|---|---|---|---|---|---|---|---|
| DC | 1-shot | 5-shot | 10-shot | Full | 1-shot | 5-shot | 10-shot | Full |
| ✗ | $\mathbf{16.9}_{\pm0.2}$ | $\mathbf{55.8}_{\pm0.4}$ | $68.2_{\pm0.2}$ | $82.5_{\pm0.3}$ | $7.0_{\pm0.3}$ | $37.1_{\pm0.5}$ | $64.1_{\pm0.4}$ | $90.3_{\pm0.2}$ |
| ✔ | $\mathbf{16.9}_{\pm0.2}$ | $\mathbf{55.8}_{\pm0.4}$ | $\mathbf{68.4}_{\pm0.2}$ | $\mathbf{82.7}_{\pm0.3}$ | $\mathbf{7.1}_{\pm0.3}$ | $\mathbf{37.2}_{\pm0.5}$ | $\mathbf{64.4}_{\pm0.4}$ | $\mathbf{91.5}_{\pm0.2}$ |
| | Aircraft | | | | VGG Flowers | | | |
| DC | 1-shot | 5-shot | 10-shot | Full | 1-shot | 5-shot | 10-shot | Full |
| ✗ | $8.0_{\pm0.3}$ | $27.4_{\pm0.3}$ | $42.8_{\pm0.2}$ | $\mathbf{85.6}_{\pm0.3}$ | $55.3_{\pm0.4}$ | $85.7_{\pm0.4}$ | $92.7_{\pm0.2}$ | $96.2_{\pm0.4}$ |
| ✔ | $\mathbf{8.4}_{\pm0.4}$ | $\mathbf{27.5}_{\pm0.2}$ | $\mathbf{43.1}_{\pm0.4}$ | $\mathbf{85.6}_{\pm0.2}$ | $\mathbf{55.6}_{\pm0.4}$ | $\mathbf{86.2}_{\pm0.3}$ | $\mathbf{93.4}_{\pm0.4}$ | $\mathbf{96.8}_{\pm0.3}$ |

**Variation of KL-divergence values between prompts and visual features $f(\boldsymbol{\theta})$.**   Here, we provide curves of KL-divergence with and without LICO. The KL values obtained by baseline w/o LICO are calculated between CE-trained visual features and fixed CLIP pre-trained vectors, and those obtained by w/ LICO are calculated between visual features and prompts learned by LICO. In Fig. 10, we can see that LICO enables stable descent of KL values, and this indicates that the LICO learned visual features indeed approach the learned language prompts. However, the values obtained by baseline without LICO fluctuate to some extent and cannot decrease further, which is attributed to the domain gap between CLIP pre-trained data and target CIFAR-10. In other words, without manifold matching, it is difficult to enable downstream visual features to be aligned with CLIP pre-trained language knowledge.

## F  Different Text Encoders

Both Word2Vec (W2V) and BERT can replace the CLIP text encoder in our LICO framework. However, they may be inferior in acting as language guidance in LICO: (1) CLIP text encoder was trained with huge amounts of image-text pairs, and the advanced architecture ViT-B/32 has stronger representation ability than W2V; (2) Even though BERT excels at representing language data, latent representation of BERT may not align with target image representations well; (3) Recent studies on prompt learning like CoCoOp has demonstrated the effectiveness of CLIP pre-trained text encoders for downstream tasks.

In Table 12, we conducted the experiments using W2V as a text encoder, mapping texts into 512-D. W2V: using fixed word embeddings and no context tokens for OT loss. W2V-P: replacing class tokens in the prompts, the context tokens are randomly initialized as Gaussian. We can see that W2V

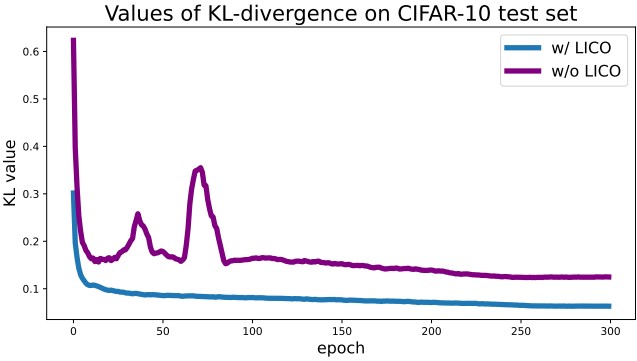

Figure 10: Variation of KL-divergence value w/ and w/o LICO on CIFAR-10 test set.

performed comparably to the None encoder baseline. W2V-P also fails, where the fixed random context tokens mislead the image features. In contrast, the CLIP outperforms both of them. The fixed W2V embedding vectors struggled to correlate well with target image features. Moreover, BERT surpassed W2V and W2V-P due to the generalizability of a stronger pre-trained model. However, BERT still cannot achieve better performances than CLIP. Furthermore, BERT-ALIGN performs competitively and even better than CLIP, which is attributed to its larger training set with more noisy image-text pairs. Consequently, from the results in Table 12 and this paper, we conclude that LICO works better with those vision-language pre-trained text encoders. Pure text encoders like W2V, even the pre-trained BERT with frozen parameters, are inferior in image-text alignment in LICO because the pre-trained parameters are not sensitive to visual features. This deficiency may be addressed by utilizing some transfer learning and domain adaptation tricks.

Table 12: Comparison of different text encoders on CIFAR-10.

| Encoder | CLIP | W2V | W2V-P | BERT | BERT-ALIGN | None |
|---------|------|-----|-------|------|------------|------|
| Full | **95.8** | 95.6 | 94.9 | 95.7 | **95.8** | 95.6 |
| 4000 | 81.5 | 81.0 | 80.2 | 81.3 | **81.7** | 80.9 |

## G  Frozen Parameters of Prompts

To evaluate the model performance with frozen parameters, we then conducted experiments on CIFAR-10 and ImageNet under two settings of frozen parameters: (1) frozen random initialization (Random) and (2) fixed form of 'a photo of a [CLS]' (Fixed). The insertion and deletion values are obtained by Grad-CAM + LICO.

In Table 13, we can see that the frozen random parameters cannot enable the models to achieve higher performances of accuracy, insertion, and deletion; the reason is that the well-trained CLIP text encoder is capable of sensing the human-understandable phrases and sentences while the random prompts lead to the difficulty in image-text alignment and yield inaccurate semantic representations. However, the form of 'a photo of a [CLS]' performs better than frozen random parameters because this meaningful prompt is more consistent with the input of the original CLIP so that the generated representations can be easily aligned with image representations.

Table 13: Evaluations on frozen parameters of prompts. "Acc." is short for "Accuracy".

| ImageNet | Top-1↑ | Insertion↑ | Deletion↓ | CIFAR-10 | Full, Acc. | 4000, Acc. |
|----------|--------|-----------|-----------|----------|-----------|-----------|
| Random | 75.88 | 53.3 | 17.8 | Random | 94.9 | 79.5 |
| Fixed | 76.20 | 55.2 | 17.4 | Fixed | 95.4 | 80.7 |
| Learnable (**ours**) | **76.27** | **57.1** | **15.1** | Learnable (**ours**) | **95.8** | **81.5** |

