# OpenReview forum: "LICO: Explainable Models with Language-Image COnsistency"
_NeurIPS.cc/2023/Conference — NeurIPS 2023 poster_

### Official Review · Reviewer_r2cD · 2023-06-29

**Soundness:** 4 excellent
**Presentation:** 3 good
**Contribution:** 3 good
**Rating:** 6
**Confidence:** 4

**Summary:**

This paper proposed LICO, which leverages the textual and semantic knowledge learned by large language models to guide latent image features. By matching relationships among images with KL-divergence globally, and distances between specific feature maps and prompt tokens with OT, the feature space of image is aligned with the prompt tokens in LLM.

**Strengths:**

1. It provides a novel idea that uses the semantic knowledge of LLM to guide the feature representations in the image classification model. Since the feature expression ability of the model is improved, both the classification performance and interpretation maps generated by XAI methods are better.
2. Thorough experiments are conducted on multiple datasets and show the effectiveness of the proposed method.
3. The paper is clearly written and easy to follow.

**Weaknesses:**

1. My main concern is that the model trained with the help of LLM is not the original model, so the improved interpretation is for the later model which has a better performance. So I'm unsure whether this can be regarded as "enhancing existing visual interpretation methods." since the interpreted model is already changed. It's more like LICO enhances the classification model so that the XAI results are improved. LICO is not an XAI method itself, but it helps the model better learn latent features by introducing knowledge from LLM.

2. Some small problems:
- What’s the transport cost used in OT?
- Since fixed promotes are in the original CLIP but learnable X1 to Xm-1 are added in the language model input, can the model work well with frozen parameters and without fine-tuning?  How are the learnable prompts initialized?
- Missing introduction in Sec.1 and Sec.2 for perturbation-based explanation methods like RISE, which is also compared in the experiments.
- The legend for g and f in Figure 2 are reversed.

**Questions:**

see the "Weaknesses"

---

> ### Author Rebuttal · Authors · 2023-08-09
>
> We appreciate your thorough summary, encouraging feedback, and constructive suggestions.
>
> **Q1:  Concerns that the model trained with LICO is not the original model**
>
> Your understanding is correct, and we completely agree with you. The proposed LICO is not strictly an XAI method but offers a more explainable classification model from scratch while maintaining its discriminative capacity. In other words, LICO acts as a *plug-and-play* training strategy, where the resulting models are enhanced with semantic knowledge of LLM through alignments of global manifolds and local features. We will make it clear in the future version.
>
> **Q2: On the transport cost used in OT**
>
> We would like to clarify that the transport cost used in OT is cosine distance, which will be clarified in the future version.
>
> **Q3: On the learnable prompt initialization and model performance with frozen parameters**
>
> Thank you for your insightful comment. We would like to note that the learnable prompts in our LICO are randomly initialized according to Gaussian distribution, the same as the setting in [1].
>
> To evaluate the model performance with frozen parameters, we then conducted experiments on CIFAR-10 and ImageNet under two settings of frozen parameters: (1) frozen random initialization (Random) and (2) fixed form of 'a photo of a [CLS]' (Fixed). The Insertion and Deletion values are obtained by Grad-CAM + LICO.
>
> In **Table R1**, we can see that the frozen random parameters cannot enable the models to achieve higher performances of accuracy, insertion, and deletion; the reason is that the well-trained CLIP text encoder is capable of sensing the human-understandable phrases and sentences while the random prompts lead to the difficulty in image-text alignment and yield inaccurate semantic representations. However, the form of 'a photo of a [CLS]' performs better than frozen random parameters because this meaningful prompt is more consistent with the input of the original CLIP so that the generated representations can be easily aligned with image representations.
>
> **Table R1**: Evaluations on frozen parameters of prompts
>
> | ImageNet | Top-1 $\uparrow$  | Insertion $\uparrow$ | Deletion $\downarrow$ | - | CIFAR-10 | Full, Accuracy $\uparrow$ |  4000, Accuracy $\uparrow$ |
> | :---: | :---: | :---: | :---: | :---: | :---: | :---: | :---: |
> | Random | 75.88 | 53.3 | 17.8 | - | Random | 94.9 | 79.5 |
> | Fixed | 76.20 | 55.2 | 17.4 | - | Fixed | 95.4 | 80.7 |
> | Learnable (ours) | **76.27** | **57.1** | **15.1** | - | Learnable (ours) | **95.8** | **81.5** |
>
> > [1] Guangyi Chen, Weiran Yao, Xiangchen Song, Xinyue Li, Yongming Rao, and Kun Zhang. PLOT: Prompt learning with optimal transport for vision-language models. In The Eleventh International Conference on Learning Representations, 2023.
> >
>
> **Q4: On the missing introduction of perturbation-based methods and reversed legend**
>
> Thank you for your careful reading and pointing out these issues. We will certainly add the introduction of RISE in Secs. 1 and 2 and fix the legend error in Figure 2 in the future version.

---

### Official Review · Reviewer_V7Pm · 2023-07-01

**Soundness:** 3 good
**Presentation:** 3 good
**Contribution:** 2 fair
**Rating:** 5
**Confidence:** 3

**Summary:**

This paper introduces LICO, a model that aligns visual encoder to language features. This model incorporates a frozen text encoder, a trainable image encoder, a classification loss, a manifold matching loss and an optimal transport loss. Experiments shows improvements over existing interpretation methods.

**Strengths:**

1. This paper is well motivated, and the method is clearly stated in text as well as figures.
2. Quantitative results are better than baselines.
3. The experiments are comprehensive in terms of the number of datasets and baselines to compare to.

**Weaknesses:**

1. LICO obtains class label text embeddings via a text encoder pretrained with internet-scale image-text pairs, thus bringing additional information. Further, the image encoder is trainable whilst in baseline methods the image encoder are all frozen. These two aspects add up to an unfair comparison to other baselines.
1. The optimal transform loss is motivated by aligning partial regions to prompt tokens, but there's no quantitative/qualitative experiments analyzing that specific effect.
2. The architectures used in this work are generally outdated, and exact version of the architecture (ResNet18/50) are less performant ones.
3. In Figure 3, the qualitative results from LICO are not better than baselines.
4. In Table 2, the improvements are not significant.

**Questions:**

1. How would authors compare the manifold loss to a cross-entropy with softmax temperature > 1?
2. What is the necessity of the text encoder being from CLIP? Would a pure text embedding model (word2vec, bert, etc) work?
3. Would a model trained with LICO work for words that are not included in the training class label set?

**Limitations:**

The authors discuss about their limitation on L303, which is not a true limitation compared to some in the Weaknesses I listed. Further, this limitation of "some training overhead" is not discussed in the paper. In concept, such an MLP would only bring insiginificant training cost when compared to the image encoder.

---

> ### Author Rebuttal · Authors · 2023-08-10
>
> Thank you for appreciating the motivation and comprehensive experiments.
>
> **Q1: On comparison fairness**
>
> We apologize for not making it clear that our comparison to baselines is fair.
>
> - **Additional textual information**. Regarding the textual information of LLM as real-world semantic space is our assumption, motivating us to enhance target models. For downstream tasks, previous interpretation method did NOT consider this critical point, so their results are biased to limited semantic space.
> - **Learned encoders of LICO and baseline methods**.  LICO and baseline encoders are trained on the same dataset from *scratch* and used the same CE loss. Even for Grad-CAM, their target encoders are trained similarly.
>
> **Q2: Analysis of aligning partial regions to prompt tokens**
>
> This question highlights an essential aspect of our research. To the best of our knowledge, LICO is the first work to align specific tokens with correlated feature maps for better interpretation. The optimal transport plan $T^\* \in \mathbb{R}^{N \times M}$ of OT implies the relationship between $M$ prompt tokens and $N$ feature maps. The corresponding row of $T^\*$ can be treated as the weights of corresponding feature maps. However, there needs further exploration on whether the weighted sum of feature maps can represent the heatmap of specific tokens, which will be studied in the future. Notably, our ablation study in Table 5 has verified the effectiveness of OT, and MF + OT outperforms MF alone both in classification performance and quantitative interpretation.
>
> **Q3: Concerns on the simple architectures**
>
> First, we followed prior interpretation methods to use these architectures for a fair comparison. Second, LICO's design is model-agnostic and mainly focused on language-consistent interpretation, which can be flexibly extended to advanced architectures.
>
> **Q4: Results in Figure 3 and Table 2**
>
> In Figure 3, although LICO and Original cover similar rough regions, LICO helps to localize more accurate details. In addition, the corresponding insertion and deletion curves quantitatively illustrate that the highlighted regions of LICO are more sensitive to model decisions. In Table 2, unlike previous methods that compromise model discriminative performance, LICO is the first method that *simultaneously* improves the interpretation and classification.
>
> **Q5: Manifold loss vs cross-entropy with Softmax temperature > 1**
>
> This question is interesting and instructive. First, for mathemtics, $L_{\text{KL}}(A^{G}||A^{F}) = - L_{\text{CE}}(A^{G}, A^{F}) + H(A^{G})$, where $H()$ denotes entropy. When the entropy $H(A^{G})$ is a constant value, KL-divergence equals CE loss. When temperature > 1, the entropy term becomes larger, resulting in the inaccurate prompt manifold. If the entropy $H(A^{G})$ is small enough, $A^{G}$ is one-hot, so we can regard KL-divergence as an equivalent of cross-entropy. Hence, MF and CE are similar to some extent.
> Second, the MF of LICO is effective in cross-modal alignment. However, CE often works with categorical one-hot labels and is inferior in matching two distributions.
>
> **Q6: CLIP text encoder vs pure text embedding model**
>
> Both Word2Vec (W2V) and BERT can replace CLIP text encoder in our LICO framework. However, they may be inferior in acting as language guidance in LICO: (1) CLIP text encoder was trained with huge amounts of image-text pairs, and the advanced architecture ViT-B/32 has stronger representation ability than W2V; (2) Even though BERT excels at representing language data, latent representation of BERT may not align with target image representations well; (3) Recent studies on prompt learning like CoCoOp has demonstrated the effectiveness of CLIP pre-trained text encoders for downstream tasks.
>
> In **Table R1**, we conducted the experiments using W2V as a text encoder, mapping texts into 512-D. W2V: using fixed word embeddings and no context tokens for OT loss. W2V-P: replacing class tokens in the prompts, the context tokens are randomly initialized as Gaussian. We can see that W2V performed comparably to the None encoder baseline. W2V-P also fails, where the fixed random context tokens mislead the image features. In contrast, the CLIP outperforms both of them.  The fixed W2V embedding vectors struggled to correlate well with target image features.
>
> **Table R1**: Comparison between CLIP text encoder and W2V on CIFAR-10
> | Encoder | CLIP | W2V | W2V-P | None |
> | :---: | :---: | :---: | :---: | :---: |
> | Full | **95.8** | 95.6 | 94.9 | 95.6 |
> | 4000 | **81.5** | 81.0 | 80.2 | 80.9 |
>
> **Q7: Would LICO work for words not included in the training?**
>
> This comment motivates us to consider the robustness of LICO. We conducted additional experiments of zero-shot inference: training on ImageNet and inference on CIFAR-10, Flowers, and FGVC Aircraft test sets.  The inference probability is the same as that image-text matching used in CLIP. At the inference stage, we replace the class tokens of ImageNet with those of test sets, and the trained context tokens are randomly selected for constructing new prompts. In **Table R2**, we report the mean accuracy of five independent tests. Compared with CLIP trained on vast amounts of image-text pairs, our LICO obtains some performance drop on CIFAR-10 and Flowers and relatively small drop on FGVC Aircraft, because CLIP training set also does not contain large amounts of aircraft images and texts. We also fixed the pre-trained encoder and fine-tune a classification head named LICO-F, which improves the zero-shot performances.
>
> In summary, our LICO can work and achieve comparable results.
>
> **Table R2**: Comparisons of zero-shot classification accuracy (%)
> | Method | CIFAR-10 | Flowers | FGVC Aircraft |
> | :---: | :---: | :---: | :---: |
> | CLIP | **75.6** | **65.9** | **19.3** |
> | LICO | 63.8 | 55.7 | 17.2 |
> | LICO-F | 70.4 | 61.1 | 18.7 |
>
> **Q8: On the limitation**
>
> Thanks. We will discuss the Weakness you listed in the future version.

---

> > ### Comment · Reviewer_V7Pm · 2023-08-16
> >
> > I thank the authors for providing this informative rebuttals. Some of them have addressed my concerns, while there are a few I still hold further questions for:
> >
> > - On the architecture. I agree with the statement of "LICO's design is model-agnostic" in the sense of computation, however when it comes to actual results ResNet and ViT would function differently and such difference would sometimes be reflected in very different attention maps. One example is the 2nd row, Figure 4 in [1].
> > - On pure text embedding model. Word2Vec is a relatively old method because 1) it was not trained on a large-scale dataset in terms of today's standard, and 2) the model may not have sufficient capacity. In some way, the table R1 in rebuttal can be explained as the deficiency of W2V instead of text-only v.s. image-text alignment. Would be more interesting to learn about the performance when the text embedding comes from a stronger model, i.e. comparable to CLIP in terms of model capacity and training data scale. A thorough discussion regarding this point would actually help to improve the quality of this work.
> >
> > [1] https://arxiv.org/pdf/2207.09684.pdf

---

> > > ### Author Response · Authors · 2023-08-18
> > > **Concerns about Different Architectures and Pure Text Embedding Models**
> > >
> > > We appreciate your further feedback. We’d like to address your further questions point-by-point.
> > >
> > > **On the architecture.**
> > >
> > > (i) We agree with you that CNN and Transformers are different in visualizing attention maps, but they are similar in incorporating with LICO due to it only depends on the latent representations, i.e., the representations before the final classification head.
> > >
> > > (ii) LICO does not affect the calculation of attention maps (the last self-attention) in ViTs. For image encoder of ViTs, LICO can also guide the representations of class tokens through proposed $L_{\text{OT}}$ and $L_{\text{MF}}$.
> > >
> > > (iii) In this paper, we follow the previous studies of interpretation, which focused on interpretation methods based on simple CNN backbones. **LICO can effectively overcome their common difficulty of improving interpretability and classification performance simultaneously**. Hence, we will treat incorporating LICO with ViTs as our future work in that there needs more effort to investigate how to obtain more explainable self-attention of ViTs, suitable quantitative metrics for interpreting ViTs, whether there exists trade-off between interpretability and classification performance, etc.
> > >
> > > (iv) Thanks for recommending the publication [1], which is a wonderful work that utilized partial distance correlation (DC) to measure similarity of different networks. The DC is helpful for generating improved attention maps via conditioning on another network, which benefits from its beautiful properties of end-to-end optimization and measuring feature spaces of different dimensions. This work inspires us to consider the relationships among features of different models and to further facilitate interpretation studies.
> > >
> > > - [1] On the Versatile Uses of Partial Distance Correlation in Deep Learning, ECCV 2022.
> > >
> > > **On pure text embedding model.**
> > >
> > > In addition to Word2Vec (W2V), we further applied pre-trianed BERT to testify the effectiveness of LICO: (i) The pure language BERT [1], (ii) The text encoder of  vision-language BERT, i.e., BERT-ALIGN, ALIGN [2] is also a framework that aligns image and language features in latent space, which differs from CLIP in training with a noisy image-text dataset that is larger than that in CLIP. For both BERT [1] and BERT-ALIGN [2], we take learnable class-specific prompts as the inputs of text encoders.
> > >
> > > Based on the Table R1, we further provide the Table R3 as follows. We can see that BERT [1] surpassed W2V and W2V-P due to the generalizability of stronger pre-trained model. However, BERT still cannot achieve better performances than CLIP. Furthermore, BERT-ALIGN performs competitive and even better than CLIP, which attributes to its larger training set with more noisy image-text pairs. Consequently, from the results in Table R3 and this paper, we conclude that LICO works better with those vision-language pre-trained text encoders. The pure text encoders like W2V, even the pre-trained BERT with frozen parameters, are inferior in image-text alignment in LICO due the pre-trained parameters are not sensitive to visual features. This deficiency may be addressed by utilizing some transfer learning and domain adaptation tricks.
> > >
> > > **Table R3**: Comparison of different text encoders on CIFAR-10
> > >
> > > | Encoder | CLIP | W2V | W2V-P | BERT[1] | BERT-ALIGN[2] | None |
> > > | :--- | :---: | :---: | :---: | :---: | :---: | :---: |
> > > | Full | $\textbf{95.8}$ | 95.6 | 94.9 | 95.7 | $\textbf{95.8}$ | 95.6 |
> > > | 4000 | 81.5 | 81.0 | 80.2 | 81.3 | $\textbf{81.7}$ | 80.9 |
> > > - [1] BERT: Pre-training of Deep Bidirectional Transformers for Language Understanding, ACL 2019.
> > > - [2] Scaling Up Visual and Vision-Language Representation Learning With Noisy Text Supervision, ICML 2021.
> > >
> > > Based on your valuable comments and suggestions, in our future work, we will comprehensively discuss different architectures  and pre-trained models and try to unify the interpretation of CNNs and ViTs.

---

> > > ### Author Response · Authors · 2023-08-21
> > > **Experimental results with Transformer-based model**
> > >
> > > To address the major concern on the backbone, we further trained a ViT-Base-16 network on ImageNet-1k dataset, and provided accuracy, insertion, and deletion in **Table R4**. We calculated the $L_{\text{MF}}$ and $L_{\text{OT}}$ between language tokens and representations of patch tokens and class token. For attention maps, we applied Grad-CAM in LICO-trained ViT-Base-16 through calculating gradients from outputs to the last attention layer of class token. **Table R4** further confirms that the transformer model with LICO not only **performs better** but also **gains better interpretability** than the one without LICO, which is in line with the finding for CNN-based backbone.
> > >
> > > In our future work, we will further develop more explainable decision clues for ViT by incorporating knowledge of LLMs into self-attention.
> > >
> > > **Table** **R4** Evaluation on ImageNet-1k using ViT-Base-16
> > >
> > > | ViT-Base-16 | Accuracy$\uparrow$ | Insertion$\uparrow$ | Deletion$\downarrow$ |
> > > | --- | --- | --- | --- |
> > > | w/o LICO | 77.9 | 55.2 | 14.4 |
> > > | w/  LICO | **78.2** | **56.0** | **13.8** |

---

> > > ### Author Response · Authors · 2023-08-21
> > > **We are looking forward to your feedback**
> > >
> > > Dear reviewer V7Pm,
> > >
> > > Thanks again for all of your constructive suggestions, which have helped us improve the quality and clarity of the paper!
> > >
> > > Since the author-reviewer discussion period will end soon in a few hours, we appreciate it if you take the time to read our further response and give us some feedback. Per your two major concerns, **we have demonstrated the effectiveness of ViT and BERT in LICO framework**, and most importantly, **ViT with LICO also obtains improved classification performance and interpretability**.
> > >
> > > Please don't hesitate to let us know if there are any additional clarifications or experiments that we can offer. If our response resolves your concerns, we kindly ask you to consider raising the rating of our work.
> > >
> > > Thanks for your time and efforts!
> > >
> > > Best,
> > >
> > > Authors of Paper 6636

---

> > > > ### Comment · Reviewer_V7Pm · 2023-08-21
> > > >
> > > > I appreciate authors' feedback and think they have addressed my concerns. I'd like to raise my score. Please ensure these improvements will be included to enhance this work.

---

> > > > > ### Author Response · Authors · 2023-08-21
> > > > >
> > > > > We appreciate your feedback and will definitely include them in the next version.

---

### Official Review · Reviewer_rFY8 · 2023-07-05

**Soundness:** 3 good
**Presentation:** 3 good
**Contribution:** 3 good
**Rating:** 5
**Confidence:** 4

**Summary:**

Most visualization interpretation methods based on saliency information often generate inaccurate saliency maps due to the limited discriminative information provided by one-hot labels. The manuscript proposes a language-image consistency model (LICO) to address this challenge. LICO utilizes a large-scale vision-language model CLIP to improve existing CAM-based vision explainable methods such as Grad-CAM. The authors assume that numerous image-text pairs used in CLIP can encode rich semantic information that can be aligned with the latent space of image domain, which establishes global manifold structure alignment and assigns local feature maps with class-specific prompts to generate more accurate saliency maps. The paper is well-organized and motivated, and the idea of leveraging language information from large VLMs is intuitive and effective. Experiments including deletion and insertion tests, sanity checks and classification demonstrate that LICO can achieve improvements over existing interpretation methods, resulting in more explainable attention maps.

**Strengths:**

(1)	The paper is well-organized and easy to understand.
(2)	The motivation behind leveraging language information from large VLMs is clear and effective.
(3)	The use of prompt information and multi-modal techniques on explainable methods may be with good value.

**Weaknesses:**

(1)	The effectiveness of LICO requires further verification. The sensitivity of LICO's saliency to model parameters raises concerns about its locality and uncertainty. Evaluations of more complex images and assessing the model's robustness would be beneficial.
(2)	The results of the saliency map are not sufficient to support the conclusion of better coverage of comprehensive and discriminative regions. Comparisons with more complex multi-target and multi-class images are preferable.
(3)	The details of the ablation studies are not clearly presented. The saliency maps generated by CAM-based methods + LICO may be insufficient to qualitatively show the salient regions that the classification model focuses on. It would be better to additionally provide results of “multi-objects single-class” and “multi-class”, except for “single-object”. Moreover, it would be better to conduct additional experiments to evaluate the segmentation and localization performance of LICO for a more comprehensive comparison with other explanation methods.
(4)	Some basic experiment settings, such as the datasets and base backbones, are expected to be explained clearly.

**Questions:**

(1)	The total loss function in algorithm 1 is inconsistent with that in Eq. (8). The Loss ‘LKL’ in algorithm 1 may need to be changed to ‘LMF‘.
(2)	Table 2 shows a decrease in classification accuracy for the base model ResNet when combined with GCC and CGC. The previous work cited suggests that good explanatory methods often sacrifice discriminative abilities. It would be helpful to explain how the proposed methods exhibit both good explanatory and discriminative abilities.
(3)	In Table 5, experimental comparisons based on Lce should also be carried out in order to better investigate the effectiveness of ‘LMF’ and ‘LOT’.
(4)	What is the effect of MLP used in the text encoder? Since the predicted probability only relies on the trained image encoder and classifier during inference, does there exist another more efficient way of processing the language features?

---

> ### Author Rebuttal · Authors · 2023-08-09
>
> Thank you for appreciating the motivation and the good value of this work.
>
> **Q1: Complex images evaluation & model robustness**
>
> Great feedback. In addition to Fig. 3, we have provided more results in Figs. 1 and 2 of the Supplementary Material (SM). For example, in Fig. 1(c) of SM, LICO captures more details, such as foot and head, and in Fig. 2 of SM, the attention maps of LICO are more explainable for human vision.
>
> For robustness assessment, we evaluated the standard deviations of the Pointing Game (**Table R1**) of different methods. LICO exhibits better results than baseline methods, indicating that LICO is more stable in localization. We will add more attention maps of complex images and assessments to the future version.
>
> **Q2: Experiments under settings of 'multi-object single-class' and 'multi-class'**
>
> We appreciate your insightful comments and suggestions. Regarding the **multi-object single-class** cases, we would like to draw your attention to Section 1 of Supplementary Material for the results. Furthermore, Fig. 2 also demonstrates LICO's capability to capture multiple entities of identical classes through its attention maps.
>
> For **multi-class** setting, we conducted the pointing game on MS COCO 2017 validation set for localization evaluation. Following settings in Score-CAM and Group-CAM, we quantified localization by calculating $\frac{\text{Hits}}{\text{Hits} + \text{Misses}}$, assessing if salient pixels fall within the annotated bounding boxes. **Table R1** shows that LICO consistently improves all the baseline interpretation methods, indicating the effectiveness of regularization by the proposed manifold OT losses. We will certainly add these results in our future version.
>
> In the **Global Response PDF** file, we provided more attention maps under ‘multi-object single-class’ and ‘multi-class’ conditions.
>
> **Table R1**: Mean accuracy values $\pm$ standard deviation of Pointing Game on MS COCO 2017 val dataset
>
> | Methods | Grad-CAM | Grad-CAM++ | RISE | XRAI | Score-CAM | Group-CAM |
> | :---: | :---: | :---: | :---: | :---: | :---: | :---: |
> | w/o LICO | 56.7$\pm$0.225 | 57.2$\pm$0.227 | 54.3$\pm$0.205 | 55.1$\pm$0.232 | 51.0$\pm$0.211 | 57.5$\pm$0.202 |
> | w/  LICO | **56.9$\pm$0.221** | **58.1$\pm$0.215** | **55.2$\pm$0.201** | **56.7$\pm$0.229** | **52.5$\pm$0.205** | **58.2$\pm$0.197** |
>
> **Q3: More details and basic experimental settings**
>
> We will add more details about ablation studies and experimental settings to the future version.
>
> **Q4: Inconsistent total loss function between Algorithm 1 and Eq. (8)**
>
> Thank you for pointing out this issue. The loss ‘LKL’ in Algorithm 1 should have been ‘LMF‘, which will be fixed in the future version.
>
> **Q5: Explanations on how the LICO exhibits both good explanatory and discriminative abilities**
>
> Very interesting observation. Based on our assumption that the CLIP text encoder implies 'real-world' semantic space, the text embedding capability is closely associated with the corresponding image space. Hence, $L_{\text{MF}}$ makes the models sensitive to 'real-world' semantic information, guaranteeing discriminative ability towards different classes. Once the image representation is globally aligned with the real-world semantic representation, $L_{\text{OT}}$ facilitates the correlation between visual feature maps and class-specific prompt tokens. For distribution alignment by OT within one image, $L_{\text{OT}}$ leads relatively redundant visual features to those learnable context tokens, highlighting the association between key features and class tokens.
>
> One critical reason for decreased classification performance by GCC and CGC is that there are only categorical one-hot labels during training. In contrast, our LICO incorporates generalized semantic information for guiding the image classification network. On the other hand, CGC is tailored for optimizing attention maps through contrastive learning but ignores preserving the discriminative ability of feature representations.
>
> **Q6: Experimental results about  $L_{\text{CE}}$**
>
> First, we would like to clarify that the Ablation studies in Table 5 are obtained by combinations of $L_{\text{CE}}$ and manifold and OT losses.
>
> In **Table R2**, we added the results obtained by $L_{\text{CE}}$. $L_{\text{MF}}$ guarantees global manifold correlated with class texts, while only using $L_{\text{OT}}$ rarely enables local alignment between feature maps and prompt tokens, ignoring class information. We observed that $L_{\text{OT}}$ decreases the performance of $L_{\text{CE}}$. This suggests that without global manifold alignment, OT struggles to enhance the discriminative ability of target classes. The reason is that OT focuses on aligning context tokens and feature maps within an individual image. Without correlation to a class-centric manifold, OT will misalign feature maps, resulting in suboptimal performance.
>
> **Table R2**: Ablation on $L_{\text{MF}}$ and $L_{\text{OT}}$
>
> | Loss | Top-1 $\uparrow$ | Top-5 $\uparrow$ | Insertion $\uparrow$ | Deletion $\downarrow$ |
> | :--- | :---: | :---: | :---: | :---: |
> | $L_{\text{CE}}$ | 76.13 | 92.91 | 53.5 | **13.3** |
> | $L_{\text{CE}}$ + $L_{\text{OT}}$ | 75.98 | 92.92 | 56.6 | 16.0 |
> | $L_{\text{CE}}$ + $L_{\text{MF}}$ | 76.18 | 92.90 | 56.9 | 15.5 |
> | $L_{\text{CE}}$ + $L_{\text{OT}}$ + $L_{\text{MF}}$ | **76.27** | **92.99** | **57.1** | 15.1 |
>
> **Q7: Effect of MLP used in text encoder**
>
> The MLP ensures that the text representations from CLIP can be aligned dimensionally with latent visual features of various image encoders in our experiment. In practical scenarios, it is possible to find an alternate method to process the text representation, given a fixed dimension. However, we emphasize that such an MLP would only bring minor training costs while not compromising inference efficiency.

---

> > ### Comment · Reviewer_rFY8 · 2023-08-17
> >
> > Thanks for the authors' effort in providing the rebuttal, which clarified several of my concerns.
> >
> > It can be claimed, to some extents, that this submission introduces a novel concept of semantic information by effectively aligning the prompt token with the feature map, .
> >
> > However, one aspect still requires further clarification. It remains somewhat unclear whether the proposed loss function genuinely contributes to the enhanced alignment. Despite that the authors added more ablation experiments on the loss function, the experimental outcomes, notably the accuracy results, indicate that the impact of LOT and LMF on performance improvements has not been consistently effective.
> >
> > Therefore, it may be more suitable to keep the current rating.

---

> > > ### Author Response · Authors · 2023-08-17
> > > **Effectiveness of proposed loss functions**
> > >
> > > We appreciate your further feedback. We would like to address your concern about the effectiveness of proposed losses.
> > >
> > > First, we'd like to clarify again that the presented LICO aims to improve interpretation ability of CNNs while maintaining or enhancing the classification performance. This is essentially challenging as discussed in previous studies such as Grad-CAM, Score-CAM, GCC, and CGC: although improving qualitative (attention maps) and quantitative (Insertion and Deletion) results, they sacrifice the decrease of classification accuracy. As shown in Tabs. 2, 3, and 4, LICO is able to **consistently** improve classification performance even under limited data settings. Most importantly, in Fig. 1 and attention maps provided in the paper and supplementary material, LICO exhibits superior interpretation ability against baseline methods.
> > >
> > > Second, in Tab. R2, LCE + LMF + LOT consistently improves the model performance. For the results in Tab. 1, LICO obtains better insertion and deletion values than baselines, and in Tab. 2, LICO outperforms CGC and GCC in classification accuracy. In the following Tab. R3, LCE + LOT and LCE + LMF also outperform accuracy of CGC and GCC. Hence, LOT and LMF are effective in improving qualitative and quantitative results compared with baseline methods.
> > >
> > > **Table R3**: Ablation on $L_{\text{MF}}$ and $L_{\text{OT}}$
> > > | Loss | Top-1 | Top-5 | Insert. | Delet. |
> > > | :--- | :---: | :---: | :---: | :---: |
> > > | $L_{\text{CE}}$ | 76.13 | 92.91 | 53.5 | **13.3** |
> > > | CGC | 74.60 | 92.24 | 52.2 | - |
> > > | GCC | 74.40 | 92.12 | - | - |
> > > | $L_{\text{CE}}+L_{\text{OT}}$ | 75.98 | 92.92 | 56.6 | 16.0 |
> > > | $L_{\text{CE}}+L_{\text{MF}}$ | 76.18 | 92.90 | 56.9 | 15.5 |
> > > | $L_{\text{CE}}+L_{\text{OT}}+L_{\text{MF}}$ | **76.27** | **92.99** | **57.1** | 15.1 |
> > >
> > > Lastly, we highlight the significance of LICO in terms of **harmoniously bridging the gap between better interpretability and competitive classification performance**. Particularly in some real applications like medical imaging and autonomous driving, LICO pioneers an effective way of  explainable AI by applying knowledge of LLMs.

---

> > > ### Author Response · Authors · 2023-08-21
> > > **We are looking forward to your feedback**
> > >
> > > Dear reviewer rFY8,
> > >
> > > Thanks again for all of your constructive suggestions. Hope our previous response can address your concerns.
> > >
> > > Since the author-reviewer discussion period will end soon in a few hours, we appreciate it if you take the time to read our further response and give us some feedback. If our response resolves your concerns, we are wondering if you would like to re-consider the rating.
> > >
> > > Thanks for your time and efforts!
> > >
> > > Best,
> > >
> > > Authors of Paper 6636

---

### Official Review · Reviewer_tr3q · 2023-07-05

**Soundness:** 3 good
**Presentation:** 3 good
**Contribution:** 3 good
**Rating:** 6
**Confidence:** 3

**Summary:**

This paper introduces Language-Image-COnsistent (LICO) to get better interpretation for classification using the Vision-Language model. The proposed framework uses a frozen text encoder and a trainable image encoder to encode text and image information. The text is composed of several trainable prompt tokens and the text label for image classes. Then, the manifold matching (MF) loss is used to align the image feature latent space with the text feature latent space. In addition, another Optimal Transport (OT) loss is used to build a fine-grained correlation between prompt tokens and image features.

**Strengths:**

1. The added text encoder during training time can introduce new information from the text encoder to the image encoder, but won't influence the inference procedure. Therefore, the inference procedure will keep the same as conventional classification models.
2. The insertion and deletion tests are used to validate the generated model interpretations from the proposed method, and LICO outperforms previous interpretation methods such as GradCAM and RISE in most of the cases.
3. The authors conduct experiments on several image classification benchmarks. For ImageNet, LICO obtains higher accuracy than the baselines. For other benchmarks such as CIFAR and SVHN, LICO achieves better performances than the baselines using limited amounts of labels. For fine-grained benchmarks such as Aircraft, LICO also shows better performances under full/few-shot settings in most of the cases.

**Weaknesses:**

1. Did you compare the training time with the baselines? The inference time will be similar, but the training process will involve an extra text encoder. Therefore, it may take longer time to train the model and more space as well.
2. In table 6, do you have the result for 0 learnable context tokens?

**Questions:**

see weakness

**Limitations:**

yes

---

> ### Author Rebuttal · Authors · 2023-08-09
>
> Thank you for your insightful and valuable feedback, pushing us to rethink more comprehensive experiments.
>
> **Q1: Training time comparison**
>
> Thank you for pointing out this issue.  We completely agree that the training time should be compared and discussed.
>
> Per your suggestion, we reported the training time of the models with and without LICO on different datasets in **Table R1**. We can see that LICO requires more training time due to the additional forward process of the text encoder and MLP.  We argue that given the model's better interpretability and classification performance, the additional training cost is acceptable and compromised as the model training can be done offline. We will certainly add this discussion to the future version and investigate a more efficient training strategy to improve the training efficiency of LICO in the future.
>
> **Table R1**: Comparison of training time (sec. per epoch) with and without LICO
> | ResNet-50 | ImageNet | Aircraft | Flowers | - | WRN | CIFAR-10 | CIFAR-100 |
> | :---: | :---: | :---: | :---: | :---: | :---: | :---: | :---: |
> | w/o LICO | 1200 | 60 | 84 | - | w/o LICO | 80 | 160 |
> | w/  LICO | 1850 | 86 | 123 | - | w/  LICO | 146 | 371 |
>
> **Q2: About 0 learnable context tokens in Table 6**
>
> We appreciate your insightful comments. We first would like to note that  '0 learnable context tokens' indicates the setting of **no learnable parameters in prompts**, i.e., only the single class tokens are used.
>
>  We provided the performance of the model learned with 0 context tokens in **Table R2**, which are relatively worse than others. This is partly due to the facts: (1) the original text encoder of CLIP was trained with prompt engineering rather than single-class tokens, and (2) CLIP has validated the superiority of learning with prompt engineering over single-class tokens. Our results are consistent with the finding in the literature that prompt learning methods like CoOp [1] and CoCoOp [2] have also demonstrated the effectiveness of learning with trainable context tokens when adapting to downstream tasks. We will update the result of Table 6 in the future version.
>
> **Table R2**: Ablation on no. of context tokens
>
> | no. of context tokens | Top-1 $\uparrow$ | Top-5 $\uparrow$ | Insertion $\uparrow$ | Deletion $\downarrow$ |
> | :---: | :---: | :---: | :---: | :---: |
> | 0 | 75.64 | 91.92 | 54.1 | 17.8 |
> | 4 | 76.03 | 92.74 | 55.2 | 17.5 |
> | 8 | 76.09 | 92.89 | 56.3 | 16.0 |
> | 12 | **76.27** | **92.99** | **57.1** | **15.1** |
> | 16 |  76.21 | 92.87 |  57.0 | 15.8 |
> |20 | 76.14 | 92.93 | 56.9 | 15.5 |
>
> > [1] Kaiyang Zhou, Jingkang Yang, Chen Change Loy, and Ziwei Liu. Learning to prompt for vision-language models. International Journal of Computer Vision, 130(9):2337–2348, 2022.
> >
> > [2] Kaiyang Zhou, Jingkang Yang, Chen Change Loy, and Ziwei Liu. Conditional prompt learning for vision-language models. In Proceedings of the IEEE/CVF Conference on Computer Vision and Pattern Recognition, pages 16816–16825, 2022.

---

> > ### Comment · Reviewer_tr3q · 2023-08-19
> > **Response**
> >
> > Thank the authors for providing additional results. The results for 0 learnable context tokens makes sense based on the explanations. After reading all other reviewers' responses, I agree with Reviewer V7Pm that the backbone should be further discussed as ResNet50/18 are both CNN-based models. Grad-CAM can be easily adapted to ViT, therefore I also want to see if the proposed LICO can still be useful for transformer-based models. Missing discussions for ViT actually limits the scope of this approach, and the paper will be more interesting if the authors could validate the proposed method on SOTA image classification models.

---

> > > ### Author Response · Authors · 2023-08-21
> > > **Experimental results with Transformer-based model**
> > >
> > > We thank the reviewer for the further feedback. To address the major concern on the backbone, we trained a ViT-Base-16 network on ImageNet-1k dataset, and provided accuracy, insertion, and deletion in **Table R3**. We calculated the $L_{\text{MF}}$ and $L_{\text{OT}}$ between language tokens and representations of patch tokens and class token. For attention maps, we applied Grad-CAM in LICO-trained ViT-Base-16 through calculating gradients from outputs to the last attention layer of class token. **Table R3** further confirms that the transformer model with LICO not only **performs better** but also **gains better interpretability** than the one without LICO, which is in line with the finding for CNN-based backbone.
> > >
> > > In our future work, we will further develop more explainable decision clues for ViT by incorporating knowledge of LLMs into self-attention.
> > >
> > > **Table** **R3** Evaluation on ImageNet-1k using ViT-Base-16
> > >
> > > | ViT-Base-16 | Accuracy$\uparrow$ | Insertion$\uparrow$ | Deletion$\downarrow$ |
> > > | --- | --- | --- | --- |
> > > | w/o LICO | 77.9 | 55.2 | 14.4 |
> > > | w/  LICO | **78.2** | **56.0** | **13.8** |

---

> > > ### Author Response · Authors · 2023-08-21
> > > **We are looking forward to your feedback**
> > >
> > > Dear reviewer tr3q,
> > >
> > > Thanks again for all of your constructive suggestions, which have helped us improve the quality and clarity of the paper!
> > >
> > > We’d like to note that **the concern of evaluating LICO on ViTs, initially raised by reviewer V7Pm, has been confirmed addressed**. Specifically, the additional Table R3 about ViTs also verifies the major merit of LICO that simultaneously improves interpretability and classification performance, which is not shared in previous popular and widely used interpretation methods. Since the author-reviewer discussion period will end soon in a few hours, we appreciate it if you take the time to read our further response and give us some feedback.
> > >
> > > Please don't hesitate to let us know if there are any additional clarifications or experiments that we can offer. If our response resolves your concerns, we kindly ask you to consider raising the rating of our work.
> > >
> > > Thanks for your time and efforts!
> > >
> > > Best,
> > >
> > > Authors of Paper 6636

---

> > > > ### Comment · Reviewer_tr3q · 2023-08-21
> > > > **Response**
> > > >
> > > > I appreciate the additional experimental results provided by the authors and would like to increase my score accordingly.

---

### Author Rebuttal · Authors · 2023-08-10

## Global Responses with a PDF file

**Comments**:

Dear Reviewers,

We thank all the reviewers for their thorough summaries and valuable feedback. The reviewers appreciate that our LICO is novel and well-motivated (**rFY8**, **V7Pm**, **r2cD**) with good value of incorporating language prompts into explainable AI (**rFY8**, **r2cD**), the experiments are comprehensive (**rFY8, V7Pm, r2cD**) and demonstrate the effectiveness of LICO (**r2cD**), good performance (**tr3q, rFY8, r2cD**), and inference efficiency (**tr3q**), while the paper is well-organized and easy to follow(**rFY8**, **V7Pm**, **r2cD**).

We have posted detailed responses to each reviewer and deeply appreciate your further feedback on whether our responses adequately address your concerns. If you have any additional comments or questions, we will try our best to address them.

Per the Q4 of **V7Pm**, Q1 and Q2 of **rFY8**, the following **attached pdf file** provide more attention maps of complex images:

 - Figure 1:  Attention maps of single-class multi-object images.
 - Figure 2: Attention maps of multi-class multi-object images.

Best,

The authors

---

### Author Response · Authors · 2023-08-21
**Summary of rebuttal and discussion**

Dear Area Chair and all Reviewers,

We sincerely thank you for devoting your time to evaluating our submission and engaging in a valuable discussion with us. Your insights have greatly enriched our work.

1. We clarify that the main strength of our LICO is simultaneously improving the interpretability and classification performance of the target backbone model.
2. A critical concern of reviewers V7Pm and tr3q is whether the LICO works well on ViTs backbone. Per this concern, we have provided an additional experiment using the ViT-Base-16 backbone, and the results show that our LICO is also helpful for improving the interpretability and classification accuracy of ViT.
3. The experimental settings in our paper are consistently based on previous widely-adopted settings on the interpretability of CNNs like Grad-CAM. Therefore, we politely argue that though evaluating transformer-based models can expand the scope of our work, it does not affect the integrity and innovation of the entire article. However, we also agree that it is worth studying a unified approach for interpreting CNNs and ViTs in our future works.

Consequently, we believe that our responses have solved all concerns of reviewers.

Finally, we will make the following changes in the revised submission according to your suggestions, including: (i) add discussion of implementation and evaluations using ViTs (by **tr3q** and **V7Pm**), (ii) localization evaluations (Table R1, by **rFY8**), (iii) pure text embeddings (Tables R1,  R2, and R3, by **V7Pm**), (iv) evaluations on frozen parameters of prompts (Table R1 by **r2cD**).

Thanks for your time and efforts!

Best,

Authors of Paper 6636

---

### Decision · Program_Chairs · 2023-09-21

**Decision:**

Accept (poster)

**Comment:**

This paper proposes an approach for explaining the predictions of a neural network by relating the reasons for a prediction to language prompts.  Through the course of the reviewing process the authors and reviewers discussed the merits of the proposed approach, with all reviewers recommending acceptance.  The ACs find that there is not sufficient justification to overturn the unanimous recommendation of the reviewers, and remind the authors to use the comments and rebuttals to update their paper given the reviewer's feedback.